# Using machine learning and beach cleanup data to explain litter quantities along the Dutch North Sea coast

Mikael L. A. Kaandorp[1], Stefanie L. Ypma[1], Marijke Boonstra[2], Henk A. Dijkstra[1], and Erik van Sebille[1]

[1]Institute for Marine and Atmospheric Research Utrecht, Department of Physics, Utrecht University, Utrecht 3584 CS, Netherlands
[2]Stichting De Noordzee, Arthur van Schendelstraat 600, 3511 MJ Utrecht, the Netherlands
**Correspondence:** M L A Kaandorp (m.l.a.kaandorp@uu.nl)

**Abstract.** Coastlines potentially harbor a large part of litter entering the oceans such as plastic waste. The relative importance of the physical processes that influence the beaching of litter is still relatively unknown. Here, we investigate the beaching of litter by analyzing a data set of litter gathered along the Dutch North Sea coast during extensive beach cleanup efforts between the years 2014–2019. This data set is unique in the sense that data is gathered consistently over various years by many
volunteers (a total of 14,000), on beaches which are quite similar in substrate (sandy). This makes the data set valuable to identify which environmental variables play an important role in the beaching process, and to explore the variability of beach litter concentrations. We investigate this by fitting a random forest machine learning regression model to the observed litter concentrations. We find that especially tides play an important role, where an increasing tidal variability and tidal height lead to less litter found on beaches. Relatively straight and exposed coastlines appear to accumulate more litter. The regression model
indicates that transport of litter through the marine environment is also important in explaining beach litter variability. By understanding which processes cause the accumulation of litter on the coast, recommendations can be given for more effective removal of litter from the marine environment, such as organizing beach cleanups during low tides at exposed coastlines. We estimate that 16,500–31,200 kilograms (95% confidence interval) of litter are located on the 365 kilometers of Dutch North Sea coastline.

## 1   Introduction

The accelerated release of mismanaged plastic waste into the global ocean gives rise to the need for effective cleanup strategies (Ogunola et al., 2018). In order to minimize the negative impact of plastic pollution on the environment, cleanup strategies need to be optimized to target the most impacted areas while limiting the economic costs (Haarr et al., 2019; Newman et al., 2015). Recent studies indicate that plastics remain trapped in coastal zones (Koelmans et al., 2017; Lebreton et al., 2019; Kaandorp
et al., 2021; Morales-Caselles et al., 2021), with at least 77% of buoyant marine plastic debris beaching or floating in coastal waters (Onink et al., 2021b). Therefore, beach cleanups have the potential to be a highly effective mitigation measure.

In addition, the plastic concentrations found on beaches are generally higher compared to other environmental compartments such as the surface water or the seafloor (Morales-Caselles et al., 2021), making beaches favorable locations for cleanup

activities. Furthermore, by limiting the resuspension of plastic items by removal, the overall plastic concentration on the beach
decreases over time and the formation of microplastic is reduced (Andrady, 2011; Haarr et al., 2020; Lebreton et al., 2019). At
the same time, as cleanup activities generally involve a large number of volunteers, awareness of the plastic pollution problem
increases, leading to a reduction of plastic waste in the local environment (Kordella et al., 2013).

Although the benefits of beach cleanups are well known, the location and timing of these activities are often not optimized.
Haarr et al. (2019) identified accumulation zones of beached plastic using the shoreline curvature and gradient in the Lofoten,
Norway, and showed that high-accumulation areas are often missed by cleanup actions. Other coastal properties like substrate
and backshore type have been found to influence debris quantities as well (Hardesty et al., 2017; Brennan et al., 2018), with
more litter accumulating in areas with increased backshore vegetation. Additionally, physical processes play an important role
in the beaching of plastics and should be considered when selecting effective sites for beach cleanups.

However, the relative importance of the various physical processes involved and how these can be parameterized remains
so far unknown (van Sebille et al., 2020; Pawlowicz, 2020). Studies have addressed the importance of the landward wind
direction for debris accumulation rates (Eriksson et al., 2013; Critchell et al., 2015; Hengstmann et al., 2017; Moy et al., 2018),
the landward ocean circulation direction (Thepwilai et al., 2021), the role of tides (Eriksson et al., 2013; Pawlowicz, 2020)
and waves (Williams and Tudor, 2001). The spatial and temporal variability of the sources, e.g. rivers, population density and
the fishing industry, also play an important role for the accumulation of plastic on beaches (Rech et al., 2014; Critchell and
Lambrechts, 2016).

In addition to the study by Haarr et al. (2019), there are several other studies that assess the prediction or monitoring
of beached plastic items using machine learning methods. These algorithms can be useful in discovering complex relations
between environmental variables and litter concentrations. In Granado et al. (2019), a marine litter forecasting model was made
using Bayesian networks, involving various variables like wave height and period, wind velocity and directions, precipitation,
and river flows. Neural networks have been used to quantify litter categories in Balas et al. (2004) and Schulz and Matthies
(2014), and deep learning methods have been used to automatically identify debris on beaches (Song et al., 2021).

In order to make data-driven methods work, relatively large and consistent data sets are necessary, whereas most observa-
tional data is sparse. Beach cleanups and citizen science initiatives can potentially provide valuable information for scientific
studies on marine pollution (Zettler et al., 2017), as these data are based on a considerable amount of person hours. Examples
of citizen science data used in marine pollution research are e.g. Hidalgo-Ruz and Thiel (2013), where schoolchildren in Chile
documented the distribution and abundance of plastic debris on beaches, and Ribic et al. (2010, 2012), where amounts of
marine debris were measured by volunteer teams on beaches in the Pacific and Atlantic.

Here, we will build upon past data-driven studies by using an unprecedented data set obtained from beach cleanup efforts
organized along the Dutch North Sea coast between 2014–2019. The number of participants (about 14,000), person-hours
(about 84,000 hours), the length of beach sampled (about 1,400 kilometers) and the fact that all beaches sampled were similar
in substrate (sandy), make this data set unique and very appropriate to apply data-driven methods. Furthermore, a large set
of explanatory variables will be created, based on environmental conditions and modelled transport of marine litter. We will
fit a random forest regression model to the observed litter concentrations as a function of these explanatory variables, and

investigate which ones are important to explain the variability in beach litter. This allows us to investigate which variables are important predictors for the amount of litter present on beaches, to get a better understanding of marine pollution, and to increase the efficacy of beach cleanups by creating a predictive model that could aid future cleanup efforts.

## 2  Data Description and region of interest

Since 2013 the North Sea Foundation, a Dutch environmental non-governmental organisation (NGO) advocating the protection and sustainable use of the North Sea marine ecosystem, has organised the national Boskalis Beach Cleanup Tour. During this tour, every year in August, the entire Dutch North Sea Coast is cleaned up by volunteers. It is the largest cleanup campaign in The Netherlands. The tour is divided into stages along the North Sea coast. The length of each stage is between 8-10 kilometres. The midway points of all stages are plotted in Figure 1 using the black crosses.

During the first three editions (2013-2015), the tour was organised over a period of a month, with one stage per day. From 2016 on, the tour took 15 days, with simultaneous cleaning of two stages per day. One cleanup team started on the Wadden Island Schiermonnikoog (most eastern cross in Figure 1), the other team started in the southwestern province Zeeland in Cadzand (most western cross in Figure 1). On day 15, both teams met halfway in Zandvoort ($\approx 4.5°$E). The cleanups started around 10.00am and ended around 4.00pm, with total cleanup times between 4-6 hours for each stage. The volunteers were guided by cleanup teams of the North Sea Foundation, which consist of professional employees of the North Sea Foundation and trained volunteers.

At each stage, all litter present on the beach was collected in plastic bags and weighed. The weighing of the collected litter was done using analogue and/or digital scales (during the stage or at the end of the stage) and carried out by one of the members of the cleanup team. Most of the litter found was plastic (estimated percentage between 80-90% in terms of numbers). The years over which weights of collected litter are available for each stage are plotted in Figure 1 using the colored squares. For most stages, weights are available for all years, in some cases stages were added in later years. Figures with the observed amount of litter per location per year are presented in the supplementary material, Figure A1 and Figure A2.

To get an impression of the mean environmental conditions along the Dutch North Sea coast, the mean surface currents are plotted in Figure 1 using the arrows (Global Monitoring and Forecasting Center, 2021), and the mean wind speed and direction are plotted using the wind rose (Hersbach et al., 2020), all averaged over August between 2014–2019. The wind is predominantly coming from the southwest. Generally, the currents move from southwest to northeast along the North Sea coast. The effect of fresh water influxes from rivers is visible around the southern province of Zeeland ($< 52°$N). The effect of this fresh water influx can be observed over considerable distances along the Dutch coast, for example in the form of fresh water lenses travelling downstream (De Ruijter et al., 1997; Rijnsburger et al., 2021). Ricker and Stanev (2020) found that locations with high salinity gradients due to a fresh water influx can act as a barrier for neutrally buoyant particles, possibly causing accumulation of litter along these fronts. Finally, not plotted in the figure, tidal currents move along the coast to the northeast during flood tide and southwest during ebb tide.

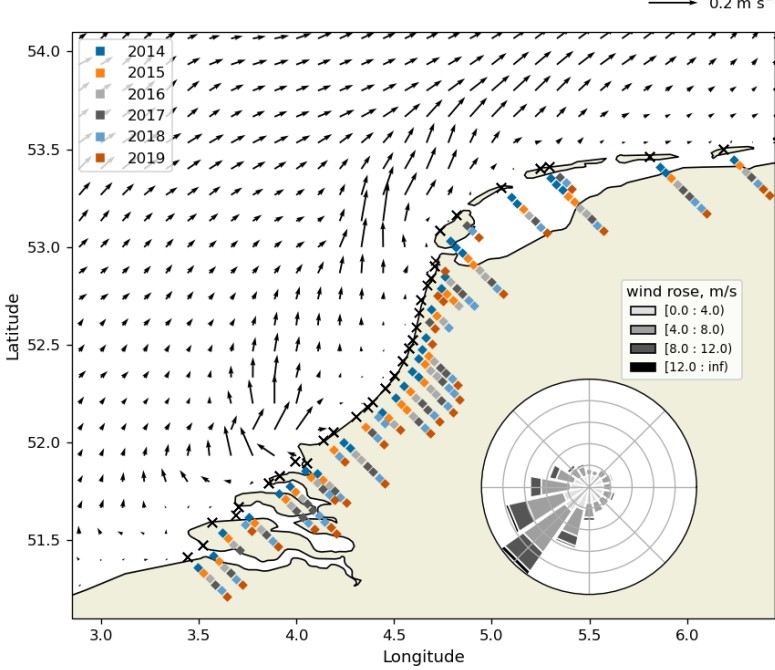

**Figure 1.** Locations of the midway points for each cleanup tour stage (black crosses); and in which year data are available (the colored squares). For stages with multiple data points per year, different stretches of beach were cleaned (e.g. once the northern side, once the southern side). Also plotted are the mean surface currents (arrows) (Global Monitoring and Forecasting Center, 2021), and the wind rose (Hersbach et al., 2020), calculated over August 2014–2019.

## 3 Methodology

### 3.1 Data preprocessing

Different sources of marine litter exist, such as mismanagement of waste near the coast, input from rivers, or fishing gear which is lost at sea. The litter is then transported through the environment, and can eventually end up on beaches, influenced by various factors such as ocean currents and winds. How all these variables combined influence the beaching of litter is unknown, however. A regression model is used here to relate various environmental variables to the observed litter concentrations. We will assess whether it is possible to use the regression model to make predictions on the amount of beached litter, and if so, which environmental variables are important predictors to take into account.

For the environmental variables, three classes of data are used. First of all, hydrodynamic data (ocean currents, ocean surface waves, tides) and wind data are used (Section 3.1.1). Furthermore, we use Lagrangian simulation data, capturing transport of virtual particles representing floating litter. These simulations are used to estimate fluxes of litter onto beaches (Section 3.1.2). Finally, we use data of the coastal geometry and orientation (Section 3.1.3). Environmental variables are calculated for various

**Table 1.** An overview of the numerical hydrodynamic and wind data used to derive the variables for the regression analysis. The data set name, temporal/spatial resolution, data used to assimilated the numerical models, and corresponding references are presented. For variables with an asterisk (*) data are used from July up to September 2014–2019. For data with a double asterisk (**) data are used for all months from January 2011 up to September 2019, as these are used for the Lagrangian model simulations as well.

| Variables | Data set name | Spatial res. | Temporal res. | Assimilated data | Reference |
|---|---|---|---|---|---|
| $U_{curr}$**, $S$* | North West Shelf reanalysis | $1/9° \times 1/15°$ | daily-mean | temperature, salinity observations | (Global Monitoring and Forecasting Center, 2021) |
| $U_{Stokes}$**, $H_s$* | Global Ocean Waves reanalysis | $1/5° \times 1/5°$ | 3-hourly-instantaneous | $H_s$ and directional wave spectra observations | (Global Monitoring and Forecasting Center, 2020) |
| $U_{tide}$*, $h_{tide}$* | FES2014 | $1/16° \times 1/16°$ | spectral | altimetry data, tidal gauges | (Lyard et al., 2021) |
| $U_{wind}$* | ERA5 global reanalysis | $1/4° \times 1/4°$ | daily-mean | various observations | (Hersbach et al., 2020) |

lead times and distances from the measurement locations (expressed as radii around the stage midway points). These variables are then fed into a random forest algorithm to make the regression model.

### 3.1.1 Hydrodynamic and wind data

Numerical model data are used to specify the state of the sea and wind around the beach cleanup locations, as these factors have been found to likely play a role in the accumulation of beach litter (Eriksson et al., 2013; Thepwilai et al., 2021; Williams and Tudor, 2001). Reanalysis data are used, where historical observational data have been assimilated in numerical models.

Information on the ocean surface currents ($U_{curr.}$), salinity ($S$), Stokes drift ($U_{Stokes}$), and significant wave height ($H_s$) are derived from E.U. Copernicus Marine Environmental Monitoring Service Information data. High frequency tidal forcing has been used to produce the ocean current data, but output is only provided daily. To capture the effects of tides on a high temporal resolution, FES2014 data are used. Tidal currents ($U_{tides}$) and heights ($h_{tide}$) are calculated, taking the $M_2$, $S_2$, $K_1$, and $O_1$ constituents into account (Sterl et al., 2020), as well as the $M_4$ and $M_6$ components which have been shown to play an important role in transport of suspended particles in the North Sea (Gräwe et al., 2014). The wind velocity field at 10m ($U_{wind}$) is taken from ERA5 reanalysis data. ERA5 data are used for the atmospheric forcing in the North West Shelf reanalysis product from which the surface current data are obtained, making these data sets consistent. Further details on the temporal/spatial resolution and assimilated data are given in Table 1.

### 3.1.2 Lagrangian model setup

While data on the sea state and wind might explain the litter accumulating on beaches to some extent, it misses information on possible sources of litter, and how this litter is transported through the marine environment. We therefore include estimates of beached litter fluxes in our analysis based on Lagrangian particle simulations.

Using the OceanParcels Lagrangian ocean analysis framework (Delandmeter and van Sebille, 2019), we model the trajectories of virtual buoyant particles at the sea surface using a Runge-Kutta 4 integration scheme. These virtual particles represent floating litter such as plastics. For the trajectories we consider a domain between $20°$W–$13°$E, and $40°$N–$65°$N, see Figure 2. We simulate a total of about 380,000 trajectories over the years 2011–2019. When particles move out of the specified domain they are removed, which mainly happens after particles move northward along the Norwegian coast. The ocean surface currents and Stokes drift from the hydrodynamic data are used to move the virtual particles around. We do not add additional tidal forcing to the Lagrangian model (Sterl et al., 2020) since the net effect of tides is already included in the ocean surface current data set (Global Monitoring and Forecasting Center, 2021). It is assumed that particles move just below the surface water, and do not experience a direct wind drag (Lebreton et al., 2018; Macias et al., 2019; Kaandorp et al., 2020). Effects of subgrid-scale phenomena are parameterized using a zeroth-order Markov model (van Sebille et al., 2018). The tracer diffusivity is set to a constant value of 10 m$^2$/s, appropriate for the given mesh size (Neumann et al., 2014).

We use the same approach as in Kaandorp et al. (2020) to define sources of marine plastic litter. Particles are released daily at river mouths, proportional to the estimated monthly riverine outflow of plastic waste based on the model by Lebreton et al. (2017). These sources are plotted using green circles in Figure 2. Particles are released daily in the sea, proportional to the amount of fishing hours based on Kroodsma et al. (2018), shown in blue in Figure 2. These data are dependent on fishing vessel transponders, which are not equally present over the years. We therefore release a constant input of virtual particles from this source each day. Finally, there is a constant daily release of particles along coastlines, proportional to the amount of estimated land-based mismanaged plastic waste within a radius of 50km from the coastline (Jambeck et al., 2015; SEDAC et al., 2005). These sources are plotted in red-brown in Figure 2.

A beaching time scale $\tau_{beach}$ parameterizes how quickly litter moves from the sea onto the beach when residing near the coast (Kaandorp et al., 2020). Here, the probability of beaching $P_{beach}$ is given by:

$$P_{beach} = 1 - e^{-t_{coast}/\tau_{beach}}, \tag{1}$$

where $t_{coast}$ is the time that particles spend in the model ocean cell adjacent to the coast. Various values for $\tau_{beach}$ are tested here, from $\tau_{beach} = 25$ days estimated for plastic particles and $\tau_{beach} = 75$ days estimated for drifter buoys in Kaandorp et al. (2020), to a more conservative value of $\tau_{beach} = 150$ days. While in reality $\tau_{beach}$ might vary significantly both in space and time, it is unknown how this can be best parameterized (Onink et al., 2021a). We use the Lagrangian model simulations to capture the large-scale transport of litter, and allow the regression model to pick the most appropriate value for $\tau_{beach}$ later on. Only direct pathways of litter through the surface water are considered here and resuspension of litter from beaches (Onink et al., 2021a) is ignored. Particles are tracked until they have lost more than 99% of their initial mass in the most conservative scenario of $\tau_{beach} = 150$ days. This means that particles are deleted when they have spent more than 691 days near the coast.

Each virtual particle starts with a unit mass. Each time step that a virtual particle spends near the coast, a fraction of its mass is lost due to the beaching process. This means that as $t_{coast}$ increases for a virtual particle, a fraction of its mass is lost, which is calculated using (1). For each virtual particle, we calculate where and when it loses mass due to the beaching process. These masses lost to beaching are binned in a $1/9° \times 1/15°$ beaching flux histogram for each day. These beaching fluxes are denoted

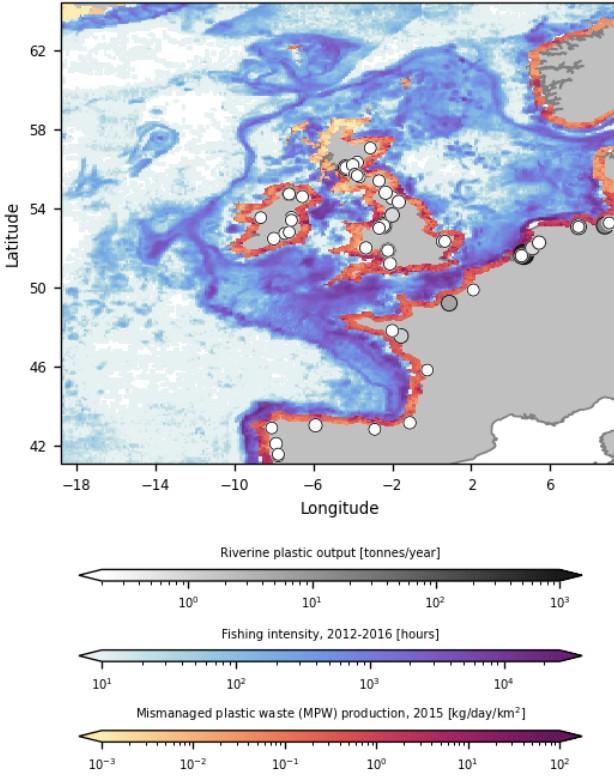

**Figure 2.** Input scenarios used to seed virtual litter particles in the Lagrangian simulations. Riverine input is indicated by the green circles, the amount of fishing hours in blue, and the coastal mismanaged plastic waste density in red. Note the log scale used for all input scenarios. While all rivers from Lebreton et al. (2017) are included in our analysis, only rivers predicted to transport more than 0.2 tonnes of plastic litter into the ocean are plotted here.

by $F_{beach}$, and are calculated for each particle source: $F_{beach,fis.}$, $F_{beach,riv.}$ and $F_{beach,pop.}$ for fishing activity, river inputs, and mismanaged plastic waste from coastal population, respectively.

### 3.1.3 Coastal orientation and geometry

Coastal orientation, geometry, and substrate are likely to influence the amount of litter that actually beaches on coastlines (Brennan et al., 2018; Andrades et al., 2018; Hardesty et al., 2017). Although the substrate of beaches in the Netherlands is relatively similar (sandy), there are local variations in the coastline orientation with respect to the large-scale coastline. We take this into account by including information on how the hydrodynamic and wind data are oriented with respect to the local coastline.

The Natural Earth data set is used here at a 1:10 million resolution (Kelso and Patterson, 2010), which is fine enough to estimate the general orientation of the beaches on which the cleanup stages have taken place. Two locations are not present

160

165

in the coastal geometry of this data set (two man-made beaches along dams: Brouwersdam and Neeltje Jans); the coastal orientations of these locations were determined manually.

Normal vectors to the coastline (denoted by $\mathbf{n}$) are estimated by fitting a tangent plane through the points defining the coastline segments. Using a singular value decomposition we minimize the orthogonal distance between these points and the plane. All points within a box of $10 \times 10$ km centered around the stage midway point are selected (roughly the length scale of the beach cleanup tours). One example is plotted in Figure 3a, where the dotted box is the selection around the stage midway point, and the coastline segments within this box are indicated in orange. The resulting normal vector to this coastline segment is plotted using the orange arrow.

Dot products are calculated for vector fields (e.g. current velocity) with respect to the coastline normal vectors, to quantify how much a vector points on-shore (positive dot product), or off-shore (negative dot product). An example is presented in Figure 3b. At a given stage midway point, the numerical data within a certain radius are selected. For each of the cells we can then calculate the dot product of the vector data with respect to the coastline normal vector. In the example of Figure 3b, the normal vector points towards the northeast. Cells where the velocity vector points in roughly the same direction (on-shore) are colored red, the opposite directions (off-shore) are colored blue. In Figure 3b the example is presented for only one time snapshot: the quantities can be calculated for various lead times. We then save derived quantities such as the mean, maximum, or minimum dot product over the lead time in a given radius, which will be further explained in Section 3.2.1.

The coastal normal vectors are also used to estimate the misalignment between the numerical model coastline and the high resolution coastline. In Figure 3a, the numerical model grid cell centers at the coast are plotted using the brown dots. A singular value decomposition is used again to estimate the coastline normal vector of the numerical grid ($\mathbf{n_{grid}}$, indicated by the brown arrow). At each stage midway point, the dot product is taken of $\mathbf{n_{grid}}$ with respect to the high resolution coastline normal vector $\mathbf{n}$, to obtain a measure for the misalignment. In the example plotted in Figure 3a there would be a large amount of misalignment between $\mathbf{n_{grid}}$ and $\mathbf{n}$, resulting in a negative dot product between the two quantities.

Finally, the coastline length per grid cell is estimated. For each cell of the numerical model, we take the coastline segments within the given cell, and calculate their total length. Since coastlines show fractal behavior (Kappraff, 1986) their Euclidian length is not well defined. This means that the lengths calculated here are estimates, and that their value would increase when taking a higher model resolution.

### 3.1.4 Spatial variability

Information on spatial variability of beached litter can be useful for cleanup campaigns to target areas which are likely to be the most polluted. One might expect that cleanup locations close to each other show more similar litter concentrations, compared to locations that are further apart. Furthermore, it is important for modelling studies to know the subgrid-scale variability which is not captured by the (discrete) numerical data (Kaandorp et al., 2020). Finally, observing how spatial variability changes for different length scales could give us clues which physical processes are important for the dispersion of litter.

We will quantify the spatial variability of litter found on the coast as a function of the separation distance between the different cleanup locations using an empirical variogram. To compute the empirical variogram, all pairs of measurements

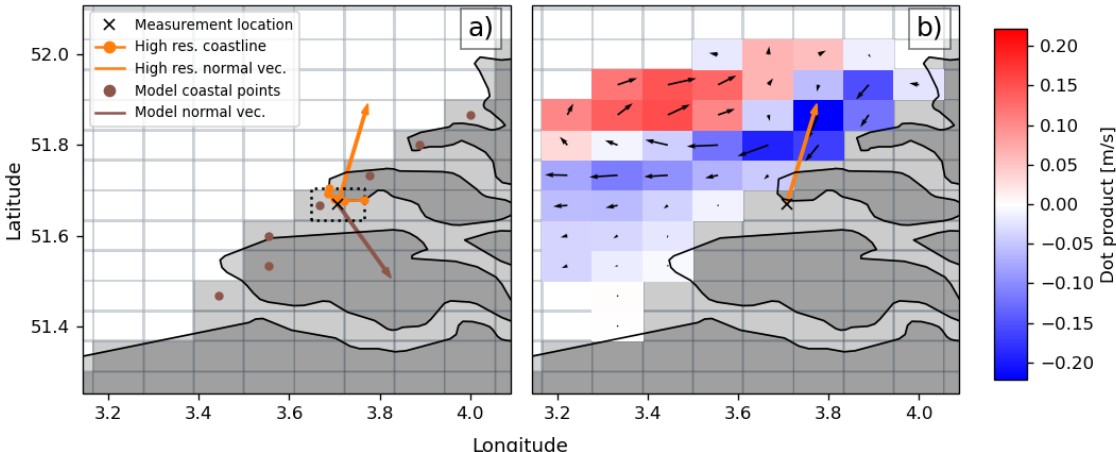

**Figure 3.** Illustration of the methodology used to calculate the directional variables. In the left panel (a), we show the high resolution coastline points and the derived normal vector ($\mathbf{n}$) in orange, located around the stage midway point (the black cross). Also shown are the numerical model coastline points and the derived normal vector ($\mathbf{n_{grid}}$) in brown. In the right panel (b), it is shown how the dot product variables are calculated. In a radius around the stage midway point, the dot product of the vector field is calculated with respect to the high resolution coastline normal vector ($\mathbf{n}$), where off-shore components are indicated in blue, and on-shore components in red.

within a certain distance of each other are compared, defined by $h \pm \delta$, where $h$ is called the separation distance, and $\delta$ is half the bin width used to discretize the separation distance. The empirical variance $\hat{\gamma}(h)$ of the measurements separated by $h \pm \delta$ is calculated using (Bachmaier and Backes, 2011):

$$\hat{\gamma}(h \pm \delta) = \frac{1}{2N(h \pm \delta)} \sum_{(i,j) \in N(h \pm \delta)} (z_i - z_j)^2, \tag{2}$$

where $N(h \pm \delta)$ denotes the number of samples in the given separation distance bin, and $z$ is the quantity of interest.

We calculate the empirical variogram on the $\log_{10}$ values of the measured plastic concentrations in kg km$^{-1}$. Confidence intervals of the calculated variogram are estimated using a jackknife parameter estimation (Shafer and Varljen, 1990).

Measured litter concentrations are subject to both spatial and temporal variability. To remove temporal variability as much as possible from the empirical variance estimates, we only use data pairs within a certain time separation. Decreasing the time separation window reduces the effect of the temporal variability, but also reduces the number of available data pairs. We use a time separation of 3 days here, for which it was found there are still enough available data pairs to compute the empirical variogram.

### 3.2 Model

#### 3.2.1 Machine learning features

The variables described in sections 3.1.1–3.1.3 are used to create a set of explanatory variables, which are related to the
observed beach litter quantities. It is, however, not obvious what kind of lead time should be considered for the variables,
and over which spatial scale the variables will have an influence on beach littering. We therefore calculate a large set of
combinations for the explanatory variables by varying the radius of influence and/or the lead time. For the radii, we will
consider the variable data closest to the stage midway point (which we will denote by a radius of 0 km), and variable data
within radii of 50 and 100 kilometers. For lead times, we will consider 1, 3, 9, and 30 days. As shown in Eriksson et al. (2013)
and Ryan et al. (2014), the turnover of litter on beaches generally happens within time scales of days, meaning that with this
range of lead times we should be able to capture most of the litter accumulation. Furthermore, a lead time of 30 days also
captures all tidal variability up to and including the spring-neap cycle. The combinations of variables, lead times, and radii will
be called features, which are fed into the regression algorithm.

An overview of the features is given in Table 2. Three categories are defined: scalar features; directional features, which contain information on the direction of various vector fields with respect to the coastline; and features derived from the Lagrangian
model simulations.

For the scalar features, we look at $H_s$, and the magnitude of $U_{Stokes}$, $U_{wind}$, $U_{curr.}$, and $U_{tides}$. We calculate the mean and
the maximum of these quantities using all data points within the given radii and lead times.

We calculate a number of features derived from the tidal height $h_{tide}$. First of all, the maximum tidal height and the standard
deviation of the tidal height over the given lead times are calculated, taking the closest data point from the stage midway point.
Furthermore, a quantity is defined giving information in which period of the spring-neap tidal cycle the stage was monitored
($h_{tide,deriv.}$). The maximum tidal height at the stage day, and the maximum tidal height at the given lead time are calculated.
We calculate the temporal derivative by subtracting both values and dividing by the lead time. A positive value means we
are approaching the spring tide, a negative value means we are approaching the neap tide. Since spring tides occur roughly
every two weeks, only lead times of 1 and 3 days are used for this feature. Finally, the minimum and maximum tidal height
encountered during each stage are calculated, since these might contribute to how much beach was sampled during that day.

The total coastline length within a given radius is calculated ($l_{coast}$), using the Natural Earth data set as explained in Section 3.1.3. To include possible local sources of litter, the population within a given radius ($n_{pop.}$) is included as a feature
(SEDAC et al., 2005), as well as the total fishing activity (Kroodsma et al., 2018) within a given radius ($n_{fis.}$). Additionally,
we want to include information on whether river mouths are present upstream of the cleanup stage. We use salinity ($S$) as a
proxy for this, as a low salinity will indicate a nearby river mouth. The mean and minimum salinity are calculated over the
various radii and lead times.

The number of participants for each stage is used as a feature ($n_{part.}$), to assess whether a lower percentage of litter is
captured at stages with less participants. These data are available for 2017–2019. For 2014–2016 only the total number of par-

**Table 2.** An overview of the machine learning features used. For each set of variables in each column, derived quantities are calculated such as the maximum, sum, or mean, over the given radius and lead time. Directional features are dot products of a given vector field with respect to the coastline normal vector $\mathbf{n}$. For parameters with an asterix$^*$, further explanation is given in the main text. For the last category (Lagrangian model features), the radius, lead time and the beaching time scale ($\tau_{beach}$) are all varied.

| Category | Scalar features | | | | | | Directional features | | Lagrangian model features |
|---|---|---|---|---|---|---|---|---|---|
| **Variable** | $H_s$, $\|\mathbf{U_{Stokes}}\|$, $\|\mathbf{U_{wind}}\|$, $\|\mathbf{U_{curr.}}\|$, $\|\mathbf{U_{tides}}\|$ | $h_{tide}$ | $h_{tide}$ | $l_{coast}$, $n_{pop.}$, $n_{fis.}$ | $S$ | $n_{part.}$ | $\mathbf{U_{Stokes}} \cdot \mathbf{n}$, $\mathbf{U_{wind}} \cdot \mathbf{n}$, $\mathbf{U_{curr.}} \cdot \mathbf{n}$, $\mathbf{U_{tides}} \cdot \mathbf{n}$ | $\mathbf{n_{grid}} \cdot \mathbf{n}$ | $F_{beach,fis.}$, $F_{beach,riv.}$, $F_{beach,pop.}$, $\tau_{beach}$=25,75, 150d. |
| **Quantity** | mean, max | max, std, deriv.* | max, min | sum | mean, min | - | mean, max, min | - | sum |
| **Radii** | 0,50,100km | 0 | 0 | 0,50,100km | 0,50,100km | - | 0,50,100km | 0 | 0,50,100km |
| **Lead times** | 1,3,9,30d. | 1,3,9,30d. | during tour* | - | 1,3,9,30d. | - | 1,3,9,30d. | - | 1,3,9,30d. |

ticipants per year is available. To estimate the number of participants per stage for these years, we first calculate the participant fractions per location over 2017–2019. These fractions are then scaled with the total number of participants over 2014–2016.

For the directional features, we calculate the dot product of the Stokes drift, wind, ocean currents, and tides with respect to the coastline normal vector ($\mathbf{n}$). Again, the mean and maximum are calculated, as well as the minimum, since this gives us additional information whether there have been strong off-shore components. These features are calculated for all radii and lead times. Furthermore, the misalignment of the numerical model coastline normal vector ($\mathbf{n_{grid}}$) with respect to the coastline normal vector is specified as a feature.

Finally, the total fluxes of beached litter from the Lagrangian particle simulations are given as features, from fisheries ($F_{beach,fis.}$), riverine input ($F_{beach,riv.}$), and mismanaged waste from the coastal population ($F_{beach,pop.}$). These features are calculated for different beaching time scales $\tau_{beach}$, all radii, and all lead times. The features are divided by the appropriate $l_{coast}$ corresponding to the radius, to get the estimated beached litter fluxes per unit length of coast. One benefit of adding beached litter fluxes from the Lagrangian particle simulations, is that potential sources of litter far away from the beaching location can be included. While the radius of influence for all features goes up to 100 kilometers, the Lagrangian model features can still include information from further away, since the virtual particles are tracked indefinitely as explained in Section 3.1.2.

### 3.2.2 Regression model

The features and corresponding response (the measured amount of litter in kg km$^{-1}$) are used to fit a random forest regression algorithm (Pedregosa et al., 2011). This model allows us to capture non-linear relations between the features and response. It is a non-parametric model, and does not require prior knowledge on the model structure. These are both important reasons to choose

the specific algorithm: coastal processes affecting dispersion of marine litter are highly complex (van Sebille et al., 2020), so we do not know a priori how the different environmental variables might interact, and how non-linear these interactions might be. The random forest regression model can aid in scientific knowledge discovery (Bortnik and Camporeale, 2021): it gives us Gini importances for all features (Nembrini et al., 2018). This is another reason for choosing this specific algorithm, as it provides us information which processes are important for predicting beached litter concentrations.

In total we have 342 features from all variable, radius, and lead time combinations. There are a total of 175 measured litter concentrations. The large number of features in comparison to the measurements makes it difficult to interpret the feature importance and could lead to overfitting. Therefore, k-fold cross validation is used to validate and test the model on a reduced amount of features, which are selected from a set of clusters.

Some features correlate as these are, for example, derived from the same variable, but for a different radius or lead time. However, we do not know a priori which of these radii and lead times are the most appropriate predictors for the beached litter quantities. For example, litter concentrations might be influenced by long-term processes, slowly increasing the standing stock of litter on the beach, or the concentrations could be better predicted by conditions on the day leading up to the cleanup stage. Since we do not know this, we let the algorithm select the most appropriate variables. Features which are highly correlated will be assigned to clusters. We use hierarchical Ward-linkage clustering for this, based on Spearman rank-order correlations (McCann et al., 2019; Cope et al., 2017). This way, the total set of features is reduced to 66 feature clusters. For further details and interpretation of the clusters see Appendix C.

Nested 5-fold cross validation is used for optimal feature selection from the clusters, and to assess the model performance on a test data set. In the outer loop, we use 80% of the data to train the model, and use the remaining 20% to test the model performance. This is repeated for each fold, i.e. 5 times. In the inner loop, 80% of the training data (i.e. 64% of the total data) is used to train the model, and 20% of the training data (i.e. 16% of the total data) is used to calculate the importance of the features, also repeated 5 times. Since in the inner loop none of the test data are used to train the model, we do not overpredict the model performance (Hastie et al., 2008). As all features in our regression model are continuous (i.e. there is no bias from categorical features (Nembrini et al., 2018)) we use the random forest Gini importance. After the inner loop is complete, we then select the feature with the highest Gini importance from each cluster. The random forest is trained using the selected features, and its performance is evaluated using the test data. We keep track of which features from the clusters are estimated to be the most important. The entire process is repeated 10 times, to obtain consistent feature importance estimates. A schematic of the model pipeline is presented in Appendix D.

## 4 Results and discussion

### 4.1 Regression analysis

The regression model shows reasonable correspondence with the measured litter concentrations, where the Pearson correlation coefficient (R) based on the repeated cross validation is $0.72 \pm 0.08$. A scatter plot with the measured litter concentrations on the x-axis and the predicted litter concentrations on the y-axis is shown in Figure 4. The points are colored according to their

test folds. As the 5-fold cross validation is repeated 10 times, only one realization is shown here, where every data point is plotted once.

In the same figure, the variability is shown that can be expected for length- and time scales smaller than the numerical data resolution. Using the empirical variogram, we calculate that $\hat{\gamma} = 0.08$ for lag distances of $h = 5 \pm 5$ km. This lag distance is at the lower side of the grid resolution for the numerical data (approximately 7 km for the ocean current data), so the model is not able to capture variations below this length scale. Therefore a 1:1 line is plotted $\pm$ 2 standard deviations based on this variance, as an indication of the optimal performance that can be expected. In this case, 94% of the predicted values lie inside the $\pm 2\sigma$ interval, indicating that the model is close to the optimal performance that can be expected for the given spatial and temporal resolution. It can be seen that there are two kinds of outliers in Figure 4: low observed litter concentrations not captured by the model (points in the upper left corner of the scatter plot), and high observed litter concentrations not captured by the model (points in the lower right corner of the scatter plot). This can be explained by the fact that the model is not able to capture all variability contained in the observations. As the hydrodynamic and wind data in the model have a limited resolution, subgrid-scale effects are missing (see Section 4.2). Furthermore, local point sources of litter (both spatially and temporally, e.g. shipping container accidents (van der Molen et al., 2021)) are not captured by the model.

In Figure 5 we show box-plots for the 10 most important features based on the Gini importance, picked out of the total 66 feature clusters. Importance scores for all 66 feature clusters are plotted in Appendix B. The model indicates that tides play an important role for predicting the amount of beached litter. The most important feature is related to the long term variability of the tidal height, with a lead time of 30 days. Short term behavior is also seen as important, as the second most important feature is the maximum tidal height encountered within a lead time of 3 days. Furthermore, the maximum tidal height encountered during the tour is the 6th most important feature, and the dot product of the tidal currents with respect to the coastline is the 8th most important feature. In general, a higher tidal maximum and variability lead to less litter measured on the coastline (see the Appendix B5 for further details). A higher tide during or preceding the cleanup could re-suspend some of the litter from the beach. Furthermore, a higher tide encountered during the cleanup stage reduces the beach width that can be sampled. Perhaps a stronger variability in the tidal height leads to less persistent high strandlines where the highest litter concentrations are normally found (Heo et al., 2013). It has been shown in numerical studies that residual tidal currents can lead to a net transport of both suspended and floating matter (Gräwe et al., 2014; Børve et al., 2021; Schulz and Umlauf, 2016). While the regression model indicates that tides play an important role, it is difficult to separate the causal relations between all these different effects and the litter quantities found on beaches. To quantify this in more detail, further experimental and numerical studies are required.

The coastline length in the neighborhood of the cleanup stage ($l_{coast}$) is ranked as the 4th most important feature. This feature can describe multiple effects on litter concentrations. More coastline per unit area means that litter concentrations are possibly spread out over longer stretches of beach, reducing the amount of litter per kilometer of beach. Furthermore, an increasing $l_{coast}$ indicates an increasing irregularity of the nearby coastline shape. This is for example the case around the province of Zeeland in the southwest ($< 52°$N in Figure 1): in these regions with irregular coastlines, more sheltered beaches can be found compared to regions with a long straight coastline, influencing the litter concentrations. Coastal orientation, $\mathbf{n_{grid}} \cdot \mathbf{n}$, plays an

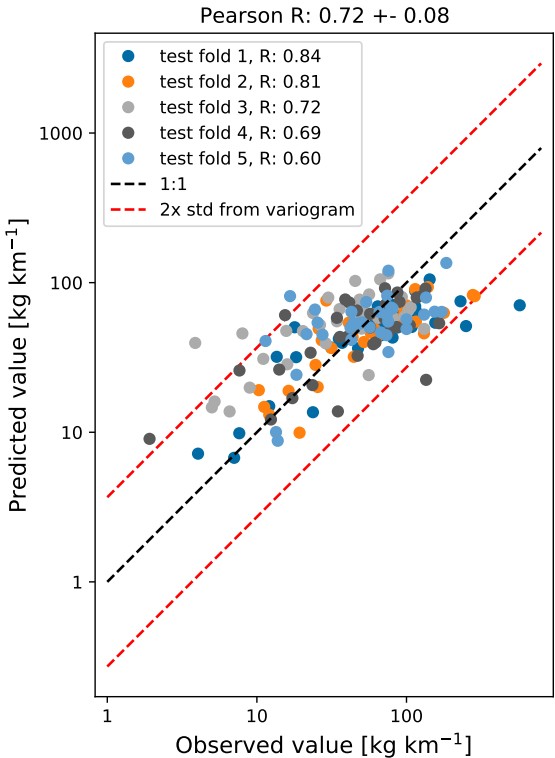

**Figure 4.** Scatter plot of the observed litter quantities (x-axis), and the modelled litter quantities (y-axis), both log-transformed. The points are colored according to the 5 test folds used in the analysis. The 1:1 line is plotted using the black dashed line, and the estimated uncertainty based on the small-scale variance ($\pm 2\sigma$) is plotted using the red dashed lines.

important role given its 5th highest Gini importance. When the coastline section tends to be more directly located towards the open sea, the large scale coastal geometry ($\mathbf{n_{grid}}$) aligns with the small scale coastal geometry ($\mathbf{n}$) at the locations used here. In e.g. Haarr et al. (2019) and Hardesty et al. (2017), it was reported that large scale headlands tend to enhance catchment of
litter compared to large scale sheltered areas. This is in line with our findings, with an increasing $\mathbf{n_{grid}} \cdot \mathbf{n}$ leading to more predicted litter (see Appendix B5).

Results suggest that transport of marine litter is important to take into account, as the 3rd and 7th most important features are beaching fluxes from the Lagrangian model simulations from fishing activity and coastal mismanaged waste. These features implicitly contain information on various hydrodynamic variables and sources of litter, explaining why these are ranked above
most other scalar and directional features related to wind, currents, and waves. Also interesting to notice, is that they are all ranked above the nearby fishing activity ($n_{fis.}$) and population density ($n_{pop.}$) , which are the 10th and 14th most important

features respectively (see Figure B1). This could indicate that transport of litter through the marine environment is important to take into account, as opposed to only considering local terrestrial sources. From the three possible sources of litter used in the model, transport from fisheries is the most important. This is consistent with the litter composition found on Dutch beaches, which consists for a major part of fishing related items (40% (van Duinen et al., 2021)).

Finally, the dot product of $U_{curr.}$ with respect to the coastline is seen as important, at place 9. This feature is related to small-scale/long-term behavior, which might give an indication whether there are currents present moving the litter on-shore to the cleanup stage location.

Changes in predictive capability are relatively small when leaving out the Lagrangian model simulation features, see Figure B2. The Pearson correlation coefficient R in this case is $0.72 \pm 0.10$, which is not significantly less than the full model. This suggests that to some extent information on transport of litter is also contained in other variables such as the currents, waves, and wind magnitude and direction. Directional information seems to play an important role, as when leaving out the Lagrangian model simulation features, 4 out of the 10 most important features are related to the dot products of currents, tides, and Stokes drift with respect to the coastline (see Figure B3).

It is estimated that the number of participants taking part in the tour does not have a large influence on the amount of litter that is found, see Appendix B for further details. This suggests that with an average of 77 participants per campaign, adding more participants would not necessarily lead to more litter being cleaned up. No clear patterns emerge regarding lead times and radii for the most important features. This could indicate that litter found on beaches is an ensemble of objects with different moments of beaching and residence times. Features regarding wind and significant wave height are seen as less important, being ranked 18th and lower, see Figure B1. It is possible that this information is already contained in the Stokes drift, or that they play a lesser role in the transport of litter. One explanation is that most of the litter found during the cleanup tour has a relatively low wind drag coefficient in the water, which was also observed in Lebreton et al. (2018) for litter in the Great Pacific Garbage Patch.

Having the full set of 66 feature clusters is not necessary for predictive capability. In Figure B4 we show that the model performs well when only picking the top 8 features (Pearson correlation coefficient R: $0.79 \pm 0.04$). Increasing the amount of features does not increase the model performance. For an operational model it would therefore be recommended to stick to a lower amount of features, as this keeps the model simple and easier to interpret. We investigate if the most important variables are related to certain locations by performing a principal component analysis, taking these 8 most important features in the full model (Figure 5). A scatter plot of the first two principal components is presented in Figure 6, where the dots are colored according to their latitude. The two principal components explain 50% and 17% of the total variance respectively. What can be seen, is that the points separate into roughly three different regions: measurements taken at lower latitudes around the province of Zeeland (51–52°N), measurements taken between 52–53°N, and measurements obtained near the Wadden Islands (53–53.5°N). The first principal component shows the highest absolute correlation (Pearson R: 0.45) with long-term tidal variability (with a lead time of 30 days). The second principal component shows the highest absolute correlation (Pearson R: -0.58) with the nearby coastal length (within a radius of 50km). As the measurements taken between 52–53°N are clustered quite closely together, this indicates that conditions regarding tides and coastline geometry are relatively similar for these

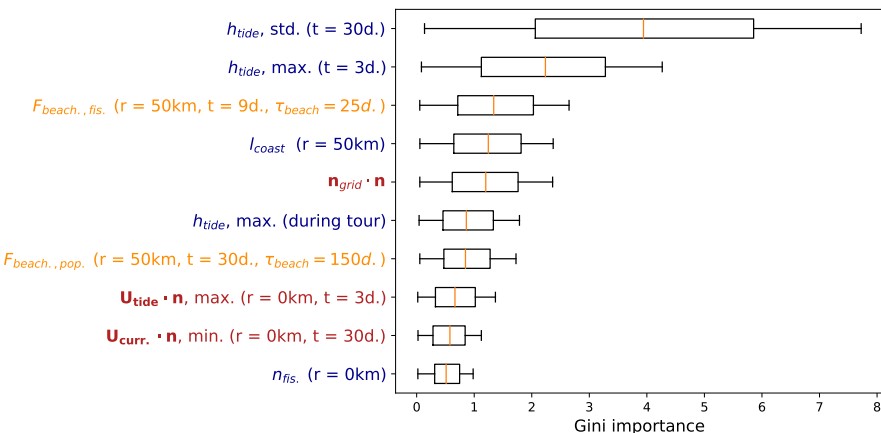

**Figure 5.** Box plots for the feature Gini importances from the random forest regression algorithm. Only the top 10 features are plotted here, an overview of all features can be found in appendix B. The label colors correspond to the variable categories in Table 2, where scalar features are indicated in blue, directional features in red, and Lagrangian model features in orange. The radius and lead time are indicated in the brackets when applicable.

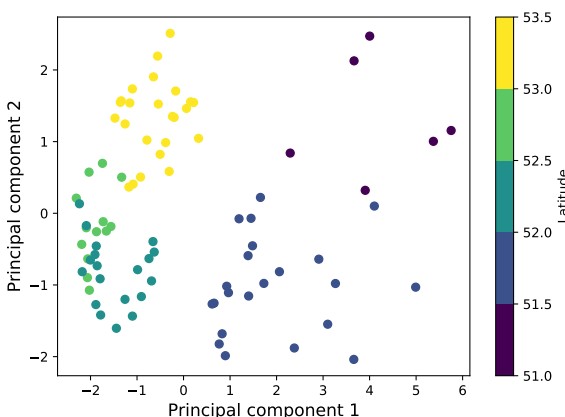

**Figure 6.** The two principal components based on the five most important features (see Figure 5). The points are colored according to their latitude, from which the separation of measurements into three different clusters (51–52°N, 52–53°N, and 53–53.5°N) becomes evident.

locations. Variations of the tidal height are relatively large for 51–52°N. The coastal geometry is also more irregular here compared to the rest of the Netherlands. These factors combined likely lead to less litter on beaches here: for $< 52°$N we find on average 52 kg km$^{-1}$, for $> 52°$N we find on average 73 kg km$^{-1}$, calculated over 2014–2019.

## 4.2 Spatial variability

To assess which length scales are important for the spatial variability of beached litter, we calculate the empirical variogram for different lag distances. Spatial variability remains relatively constant for lag distances up to about 100 km, with a mean of $\hat{\gamma} = 0.07$, see Figure 7. For the smallest lag distance ($h = 5 \pm 5$ km), we find $\hat{\gamma} = 0.08$. This variance estimate was also used to create the error bars in Figure 4. Around $h = 125$ km there seems to be an increase in the variance, to about $\hat{\gamma} = 0.2$–$0.3$. At this lag distance there is also a large uncertainty in the estimates however, and fewer unique data pairs to calculate the empirical variance.

Interestingly, some periodic behavior seems to be present, with a length scale of about 25 kilometers. One possible explanation could be the typical spacing of the Dutch islands and peninsulas. As shown in the previous section, coastline orientation likely plays an important role in the amount of observed litter. This effect can also present itself in the variogram with, for example, measurements in sheltered areas (e.g. coves) being more correlated with each other, compared to nearby exposed locations (e.g. headlands).

The grid sizes used for our numerical data ranges from about 7 km (the surface current data), to about 20 km (the wind data). This means that the variance at and below these length scales is not captured by the numerical data. The variance calculated for lag distances up to 20 km is quite substantial ($\hat{\gamma} = 0.05$–$0.12$). As can be seen in Figure 4 the values corresponding to the lower and upper 95% confidence interval vary by about an order of magnitude. This is essential to consider when using observational data to inform numerical models: due to the amount of variability at the subgrid-scale level, relatively large sets of observational data are required to extract information. A large number of physical processes could induce variability below length scales of 20 km, such as Langmuir circulations, or processes in the coastal zone such as wave breaking, rip currents, and longshore currents (van Sebille et al., 2020). Finally, it is important to consider that spatial variability is inherent to data obtained from cleanup campaigns such as analysed here, due to e.g. different participants having slightly different strategies for finding litter on beaches.

## 4.3 Extrapolating litter quantities to the entire coastline

The random forest regression model can be used to extrapolate how much litter is likely beached along the entire Dutch coastline. First, a regression model is trained using the top 8 features listed in Figure 5. We then divide the Dutch North Sea coastline into $1/9° \times 1/15°$ sections (roughly 7 by 7 kilometers). For each of the sections the top 8 features are computed, as well as the total coastline length contained in each section. In total we have 65 separate sections, and a total coastline length of 365 kilometers, which matches the total length of the Dutch North Sea coastline from literature (Roomen et al., 2008). We choose to use a model trained using the top 8 features for the extrapolations, as increasing the amount of features does not increase the predictive performance (see Figure B4). Furthermore, reducing the amount of features simplifies the computations, as we do not need to compute all 391 variables again for all coastline sections.

For each section, the litter concentrations in kg km$^{-1}$ are predicted per day over August 2014–2019. Predictions are only made for August since all cleanup campaigns were organized during this period, and making predictions for other months

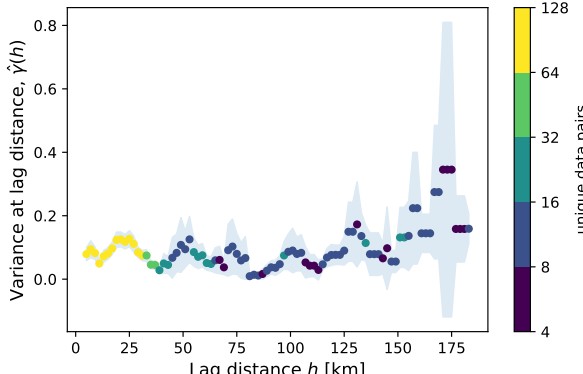

**Figure 7.** Variogram calculated for the $\log_{10}$ of the measured litter quantities in kg km$^{-1}$, with the lag distance $h$ on the x-axis, and the empirical variance $\hat{\gamma}(h)$ on the y-axis, only taking data pairs into account with a maximum of 3 days temporal separation. For the separation distance half bin width $\delta = 5$ km is used. The points are colored by the number of unique data pairs used to calculate the variance, the jackknife uncertainty estimate ($\pm\sigma$) is shaded in blue.

might induce seasonal biases. The mean concentrations per coastline section are plotted in Figure 8. For each day, the total litter quantities are computed by multiplying the litter concentrations by the coastline length per section. Monte Carlo estimates

of the confidence bounds are calculated by randomly adding noise proportional to the estimated variance ($\hat{\gamma} = 0.08$), which is repeated 1000 times per day per section.

We find a total of 16,500–31,200 kg litter along the Dutch North Sea coastline based on the 95% confidence interval. It must be noted that this only accounts for the visible litter on the beach surface. The cleanup efforts are likely to miss a substantial amount of beached litter which is buried in beach sediment or located at the back of the beach (e.g. in vegetation). This was

420 for example noted in Lavers and Bond (2017) for a remote island in the South Pacific, where in terms of mass about 68% of the litter was located on the beach surface, 27% at the back of the beach in and around vegetation, and 5% buried in beach sediment. Further research is necessary to quantify how these numbers translate to Dutch beaches.

The total amount of litter gathered during the cleanup campaigns, and the total amount of kilometers sampled per year is presented in Table A1. The total amount of litter gathered varies from 9,872 to 20,078 kilograms. This is in line with the

425 expected total amount of litter predicted by the model, since the majority of the coastline (222–262 kilometers out of 365 kilometers) was covered during the cleanup campaigns.

## 5   Conclusions and recommendations

Using data from beach cleanup efforts in the Netherlands between the years 2014–2019, we analyzed which variables are important for predicting litter on beaches, and what spatial variability this litter has. In order to do this, we fitted a regression

model to the observed litter quantities, as a function of variables related to wind, waves, currents, tides, coastal geometry,

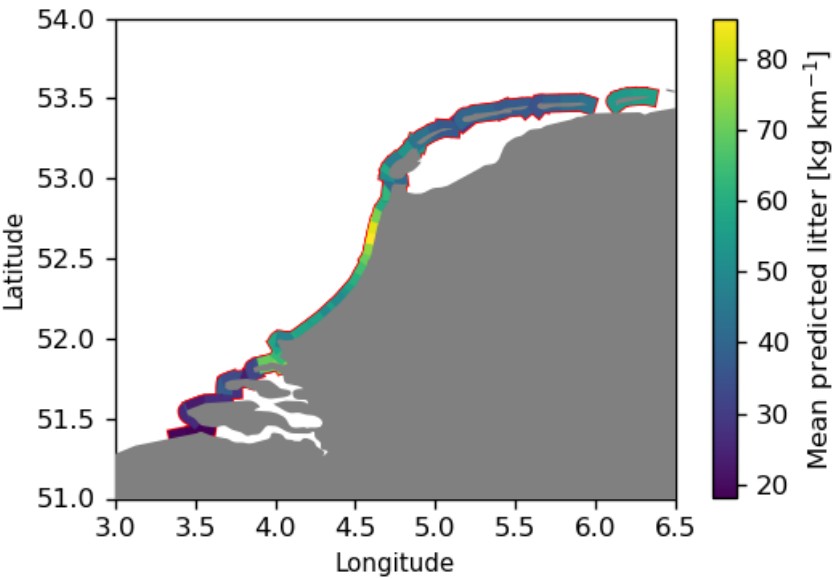

**Figure 8.** Mean litter concentrations over August 2014–2019 extrapolated to the entire Dutch coastline.

and simulated oceanic transport. We find that tides play an important role, where an increasing tidal variability and increasing tidal maximum lead to less observed litter on beaches. Other important variables are whether the local orientation of a beach corresponds to the large-scale coastline orientation, and the total nearby coastal length, which can both be seen as measures of how exposed a beach is. These factors are likely explanations why the observed litter quantities are relatively low in the south-
435 western part of the Netherlands compared to the other parts. Additionally, transport of litter through the marine environment is seen as important to take into account by the regression model. Rivers, fishing activity, and mismanaged plastic waste along coastlines were taken into account as possible sources of litter in the transport model, where the regression analysis attributed relatively much importance to litter originating from fishing activity. This is in line with findings in van Duinen et al. (2021), as approximately 40% of the litter found on the Dutch North Sea coastline is estimated to originate from the fishing industry.
We compute that spatial variability of the observed litter concentrations is substantial on length scales less than 10 kilometers, causing model $\pm 2\sigma$ confidence bounds to vary by about an order of magnitude. Due to this significant variability, large observational data sets are necessary if they are to be used to inform numerical models. Finally, based on extrapolation of the regression model, we estimate that the Dutch North Sea coastlines contain a total of 16,500–31,200 kilograms (95% confidence interval) of litter on the beach surface.
Estimating the spatial variability of beached litter can give us information for efficient monitoring of pollution. It can be used to constrain estimates of litter concentrations based on observations elsewhere. We found that the variance for lag distances smaller than 125 km is relatively constant around $\hat{\gamma} = 0.08$. As an example, if one measures a relatively high amount of 200 kg km$^{-1}$ at the northern tip of the mainland near Den Helder ($\approx 53°$N in Figure 1), one can expect at least 54 kg km$^{-1}$ of litter

elsewhere in the Northern part of the Netherlands, taking the 95% confidence interval. After 125 km, the estimated variance seems to increase, meaning that this observation becomes less informative for locations further away.

For future studies on quantifying beach litter variability, it would be interesting to segment the beach cleanup tours into smaller stretches. One idea would be to organize some stages where the litter quantities are weighed per 1 kilometer, 100 meter, or even shorter stretches. This way it would be possible to estimate the variance on sub-kilometer scale. Ryan et al. (2020) reported significant correlations between measurements taken roughly 50 meters apart (Spearman rank correlation of about 0.9). It would be interesting to see how this changes up to the kilometer scale. This can give us valuable insights into which processes might be causing the high amount of variability between litter observations, and what length scales should be taken into account to capture this variability with models. We see relatively few data points in Figure 7 for larger lag distances. Performing the cleanup stages in a randomized order would provide a more even coverage of data points over the given lag distances.

Future studies could further investigate the causal relations between the variables seen as important predictors by the regression model and the litter concentrations found on beaches. This is especially the case for tides, which constitute the two most important features in the regression model (see Figure 5). Experimental studies could further determine whether lower litter concentrations at locations with higher tidal variability are mainly caused by litter re-suspending back into the sea, or for example due to the fact that less area of the beach is sampled during high tide. It should additionally be investigated how these effects compare to the role of (residual) tidal currents, as it has been shown that this can play an important role in transporting suspended matter towards the shore (Schulz and Umlauf, 2016). Experimental investigations can be done in combination with numerical studies of the nearshore marine environment, to capture the interactions between processes such as tides, waves, and particle sizes (Alsina et al., 2020).

It should be investigated how the results found here generalize to other geographic regions, and how the importance of explanatory variables vary globally. The model itself can not directly be used for other geographic regions, since the features used to train the algorithm are specific to the region of interest. The model is likely to perform poorly when making extrapolations for conditions not present in the training data. As an example, the substrate of beaches is likely to have a large impact on litter concentrations (Hardesty et al., 2017), which are relatively uniform in this analysis (all sandy beaches). According to our regression model, wind is not a very important variable to take into account. Perhaps some of the high-windage litter has been beached before reaching the Dutch waters. It should be noted, however, that wind indirectly affects other variables such as the ocean currents and therefore also the Lagrangian particle simulations. It would be interesting to re-do this analysis with data obtained nearby the English channel and check if wind plays a more important role there, as in the Lagrangian model simulations many virtual particles pass this region.

It is necessary to further investigate the effect of regular cleaning of beaches by municipalities and other volunteer groups or individuals. This effect was left out in this analysis due to unavailability of these data. It is likely that mainly the beaches near densely populated areas are regularly cleaned. Since data on population density has been included in the features, it is possible that this effect is taken into account by the regression model, but further analysis is necessary. Furthermore, effects of tourism can be taken into account in the future when these data are available, as this affects the local population density seasonally.

Regarding effective cleanup of beaches, it is recommended to perform beach cleanups during low tide, preferably in a week around the neap tide, when the tidal variability is lower. If limited resources are available, one can focus on exposed shorelines which generally accumulate more litter. Additionally, more litter can be expected on relatively straight shorelines, compared to more irregular geometries where litter is distributed over longer stretches of beach. We saw no effect from the number of participants per beach cleanup tour on the amount of gathered litter, with an average of 77 participants per tour. One possible improvement to clean up more litter could therefore be to spread out participants over different stages, avoiding that parts of the beach are inspected multiple times.

*Code and data availability.* Code used to conduct the experiment and to create all figures, and the beach cleanup data from Stichting De Noordzee are available at https://doi.org/10.24416/UU01-NVGL3G

*Acknowledgements.* This work was supported through funding from the European Research Council (ERC) under the European Union Horizon 2020 research and innovation programme (grant agreement No 715386). Funding was provided to S.Y. by Galapagos Conservation Trust and Evolution Education Trust, Pathways to Sustainability and the K.F. Hein Fonds. The North Sea Foundation thanks all volunteers participating in the Beach Cleanup Tour. We also like to thank all sponsors and partners that make the Beach Cleanup Tour possible.

*Author contributions.* MK designed and conducted the study, with initial data analysis from EvS and steering and discussion from EvS, SY, MB, and HD. Data Curation beach cleanup tour data: MB. All authors contributed to the manuscript.

*Competing interests.* MB is employed by the North Sea Foundation. All other authors declare no competing interests

 **Appendix A:  Observational and modelled data per year**

Figure A1 and Figure A2 present the modelled litter quantities (left columns) and the raw observational data (right columns) per year per cleanup stage. The litter concentrations are plotted using circles, where the color and size correspond to the litter quantities (note the logarithmic scale here). Table A1 presents the total gathered litter per year.

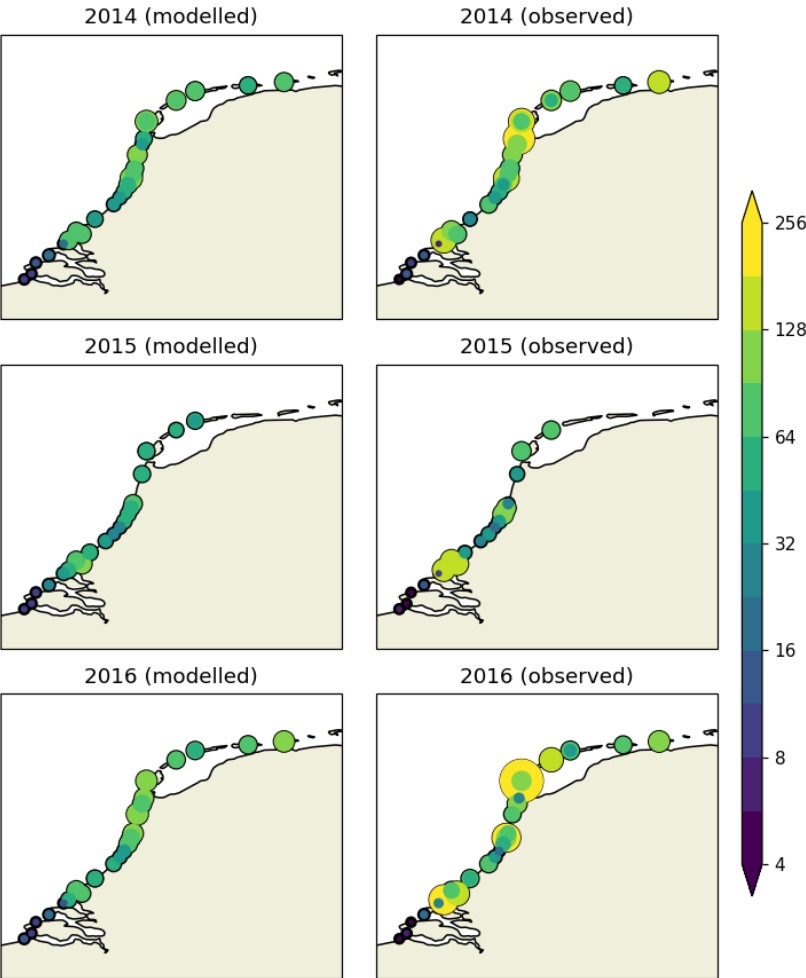

**Figure A1.** Modelled (left column) and observed (right column) litter concentrations in kg km$^{-1}$ per individual location and year (2014-2016). Circles are scaled and colored according to the litter concentrations.

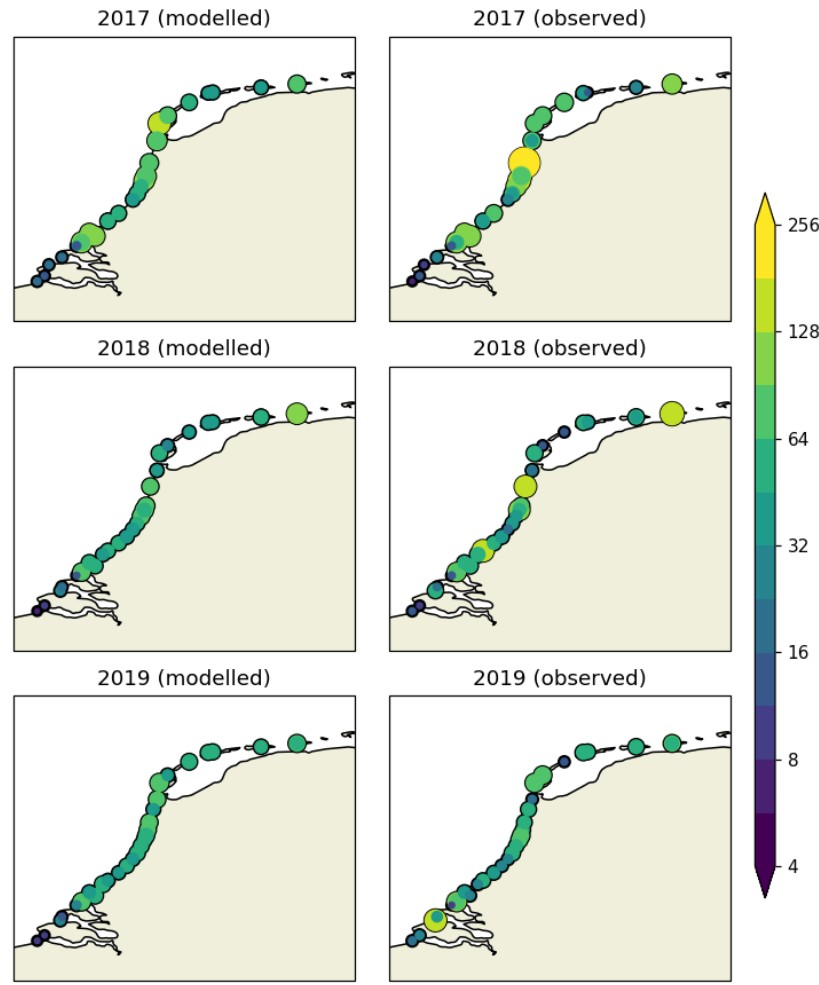

**Figure A2.** Modelled (left column) and observed (right column) litter concentrations in kg km$^{-1}$ per individual location and year (2017-2019). Circles are scaled and colored according to the litter concentrations.

**Table A1.** Overview of the total amount of litter gathered per year during the beach cleanup tours.

| Year | 2014 | 2015 | 2016 | 2017 | 2018 | 2019 |
|---|---|---|---|---|---|---|
| **Total litter gathered [kg]** | 20,078 | 9,872 | 19,203 | 14,863 | 11,163 | 10,991 |

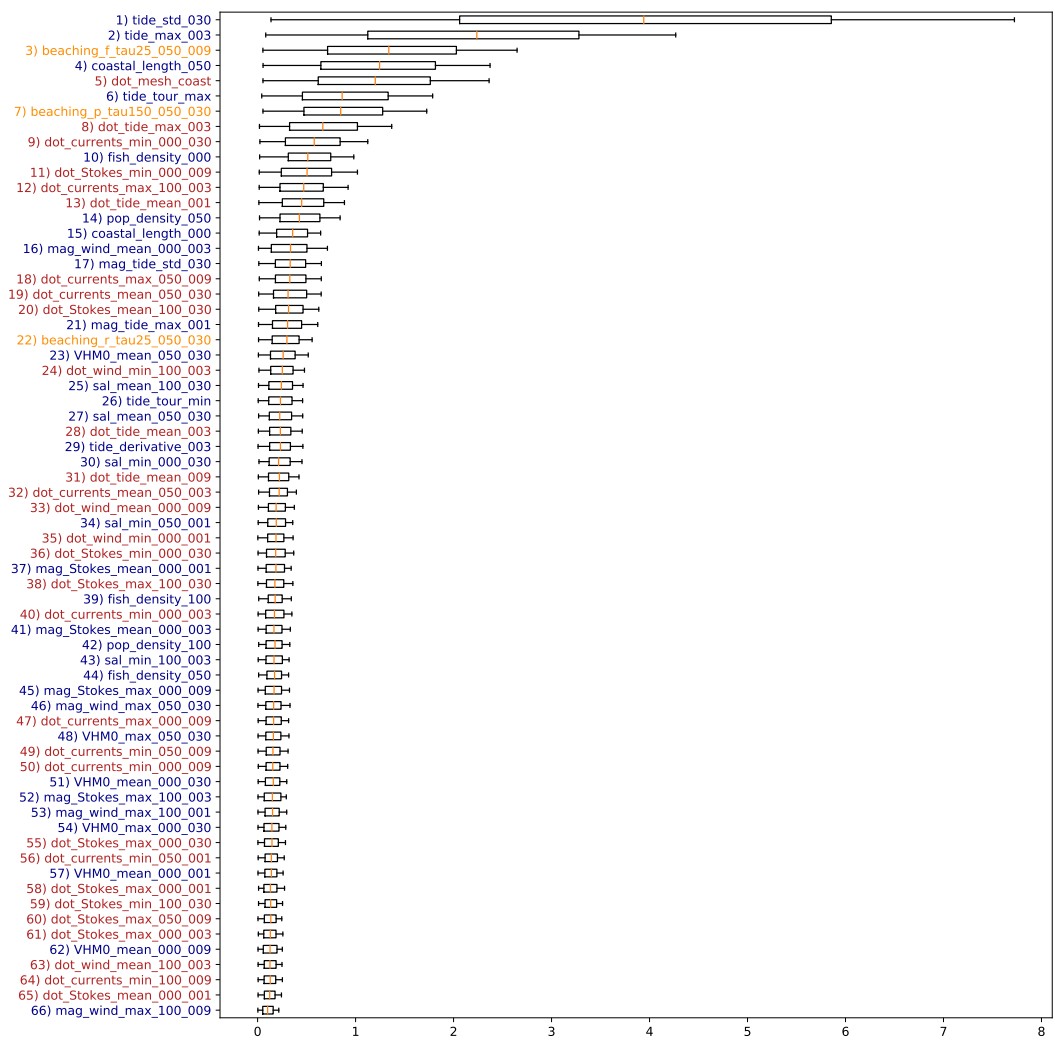

**Figure B1.** Gini importance overview of all features; again labels are colored according to the feature categories in Table 2.

## Appendix B: Extended results

### B1 Gini importance overview

A complete overview of the Gini importance for all features is presented in Figure B1. The numbers in the feature labels give information on the radius (in kilometers) and lead time (in days) if applicable, and in this order. See Table 2 for the radius and lead time combinations used for the variables. The Lagrangian model features (orange labels) are indicated by 'beaching_p', 'beaching_r', 'beaching_f', for litter sources originating from mismanaged coastal plastic waste (p), rivers (r), and fishing activity (f) respectively.

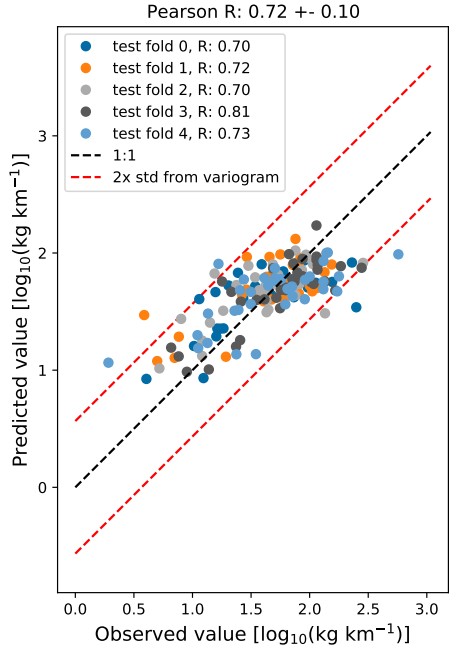

**Figure B2.** Scatter plot of the observed litter quantities (x-axis), and the modelled litter quantities (y-axis), when not taking Lagrangian model features into account. Litter quantities are log-transformed, and points are colored according to the 5 test folds used in the analysis.

## B2    Excluding Lagrangian model features

A scatter plot of the measured litter concentrations versus the predicted values is presented in Figure B2, where Lagrangian model features have been excluded from the feature set. As described in the main text, no significant decrease in the correlation is observed compared to the case where Lagrangian model features have been included ($0.70 \pm 0.10$ versus $0.71 \pm 0.11$).

The complete overview of the feature Gini importances corresponding to the case without Lagrangian model features is presented in Figure B3. As mentioned in the main text, more features related to the currents and Stokes drift orientation with respect to the coastline are seen as important now, compared to Figure B1. This could be explained due to these features taking over the role of the Lagrangian model features in capturing the effect of marine litter transport.

## B3    Effect of using only the top N features

It is not necessary to include all 66 feature clusters for predictive capability of the model. In Figure B4 we present the Pearson correlation coefficient R as a function of the number of features included in the random forest algorithm, both with and without using the Lagrangian model features. Each time only the top features (corresponding to Figure B1 and Figure B3)

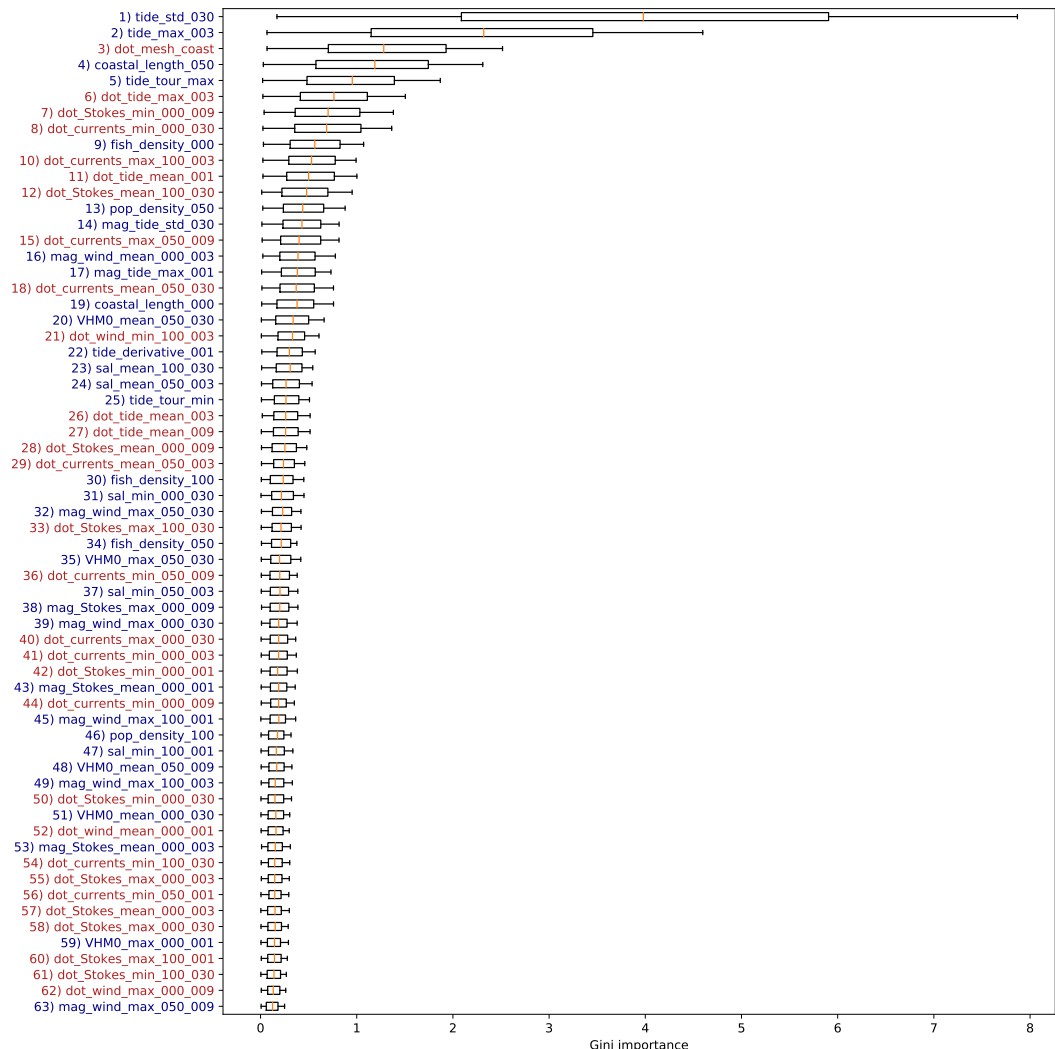

**Figure B3.** Gini importance overview when not taking into account the Lagrangian model features, where labels are colored according to the feature categories in Table 2.

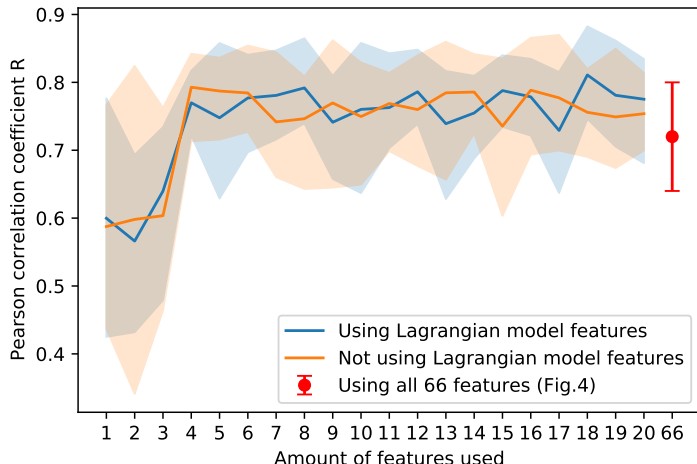

**Figure B4.** The effect of the number of included features on the Pearson correlation coefficient R, where the mean is plotted using the solid line, and the filled area represents the 10% to 90% quantile. Both the case with and without using Lagrangian model features is presented (blue and orange lines respectively). The case using all 66 features (corresponding to Figure 4) is shown using the red error bar

are used to train and test the model, using 10 times repeated 5-fold cross validation. Generally the model performs well with about 7–8 features used. Performance is quite stable in the case when Lagrangian model features are used, some outliers with
lower Pearson correlation coefficients can be observed when not taking into account these features. The Pearson correlation coefficient when using all 66 features, corresponding to Figure 4, is shown using the red error bar. In this case the Pearson correlation coefficient is slightly smaller than when using, for example, the top 8 features, which could indicate a small amount of overfitting, although this difference is not significant.

In Figure B5, we analyse the effect of leaving out certain feature categories on the model performance. The random forest
can create a highly non-linear map between the features and corresponding response. It is therefore possible that when using a large set of features and leaving out one important explanatory variable, it will use a combination of the remaining features to still obtain a good fit. We therefore only use the top 10 features in this analysis, and exclude the Lagrangian model variables, as these implicitly contain information on the other features. As can be seen, leaving out a certain category of features reduces the model performance. This can especially be observed when leaving out all features regarding tides, and the two features
regarding coastal properties ($l_{coast}$ and $n_{grid} \cdot n$). The mean Pearson correlation coefficient decreases and the variance of the model performance increases.

## B4   Number of participants

As mentioned in the main text, the number of participants is not seen as a important in terms of the Gini importance. The number of participants is correlated with the population density in the neighborhood of the stage, and is therefore assigned to

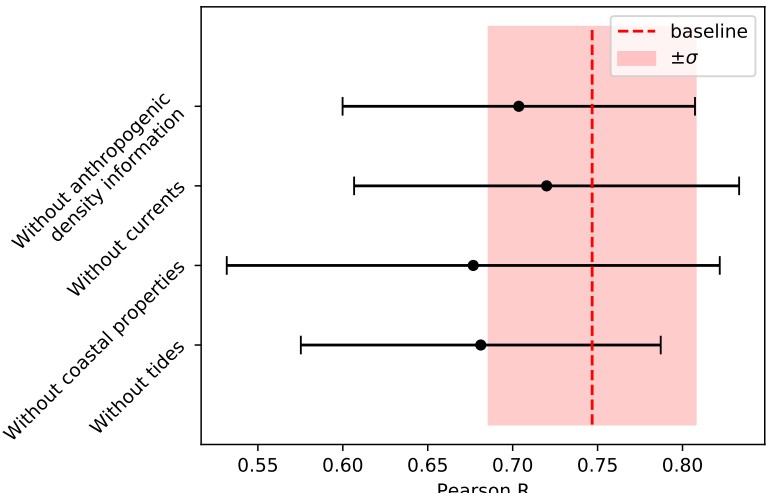

**Figure B5.** Analysis where some of the feature categories have been left out. The top 10 features have been used without the Lagrangian model features (see Figure B3, as these implicitly contain information on all feature categories. As can be observed, leaving out a set of features generally decreases the predictive performance of the model, and increases the variability of the prediction quality.

the same feature cluster as the population density, for more details see Appendix C. The number of participants was not picked out of this cluster as one of the most important features during the k-fold cross validation. In order to separate the effect of the number of participants per cleanup stage, a model run was done without the nearby population densities as features. A summary of the resulting Gini importances is shown in Figure B6, where only the top 10 features and the number of participants are plotted.

**B5   Feature effect**

The general effect of some features was described in the main text, such as the fact that an increasing tidal variability, and misalignment of the high resolution coastline with respect to the numerical model coastline ($\mathbf{n_{grid}} \cdot \mathbf{n}$) lead to less observed litter. Figure B7 illustrates this, by varying one feature on the x-axis, and plotting the resulting predictions on the y-axis. In the decision trees of the random forest, decision boundaries are made at optimal splitting locations, making the resulting model highly non-linear. This makes it difficult to interpret the regression model. In Figure B7, we 'fix' all features except the one listed on the x-axis. This feature is then varied from its minimum until its maximum encountered value. Since the random forest result can depend highly on the exact value of the other features, noise is introduced. Each other feature is varied uniformly between its 0.4–0.6 quantile, to illustrate whether the found relation for the given feature on the x-axis is robust.

Features which show relatively robust relations are related to tidal height, where an increasing variability, and a higher
maximum decrease the predicted litter concentrations. The effect for $\mathbf{n_{grid}} \cdot \mathbf{n}$ also seems to be robust, with increasing values leading to more predicted litter. For the coastal length in the neighborhood ($l_{coast}$) an increasing value seems to lead to less

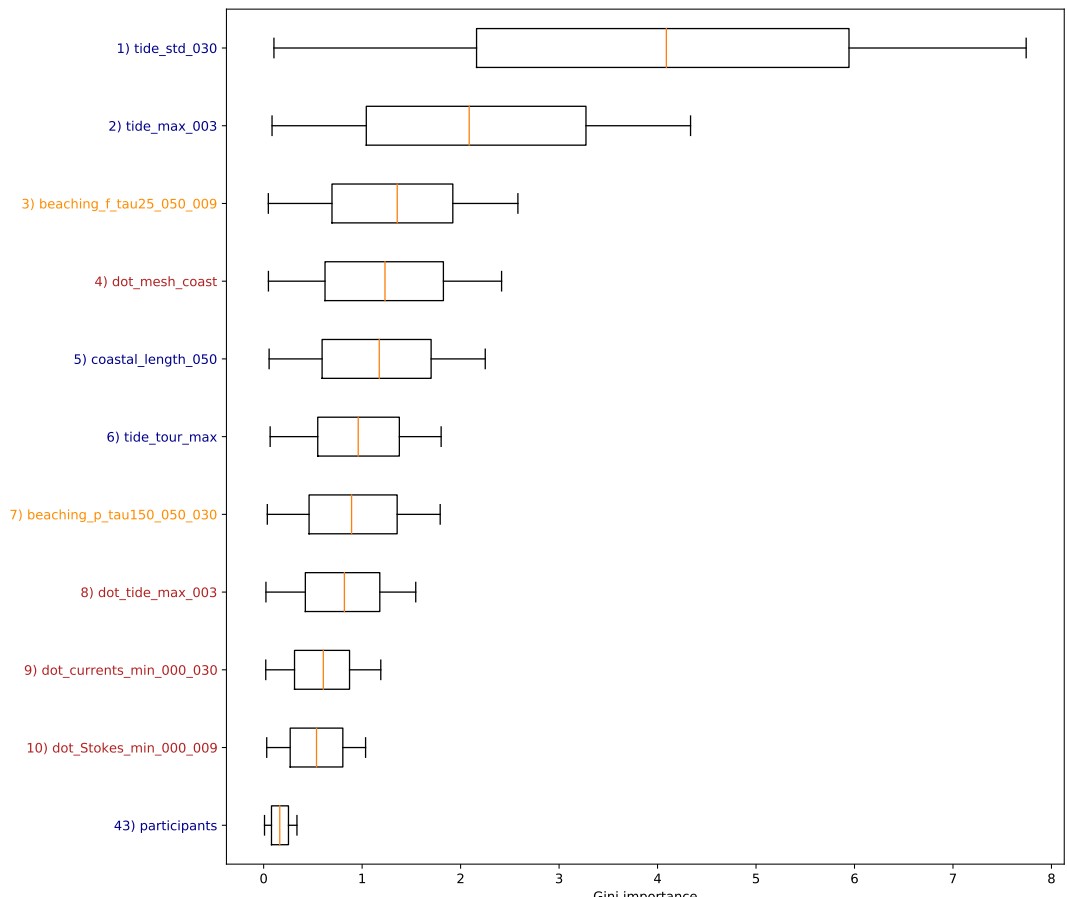

**Figure B6.** Gini importance overview when not using nearby population densities as features, to separate the effect of the number of participants per cleanup stage. In that case, it is the 28th most important feature.

litter, although there is a sudden drop observed here. This might be caused by the fact that there are relatively little data points available where this feature has a high value (most of the stages were conducted on relatively straight coastline sections), so the model has trouble learning a relation here. For the Lagrangian model features, increasing values lead to more predicted litter as expected. For the mismanaged coastal plastic waste (indicated by 'beaching_p_tau25_050_009'), the results are quite dependent on the values of other features, as a lot of noise can be seen here. Generally, the model indicates there are increasing litter concentrations for increasing currents and on-shore Stokes drift.

## Appendix C: Clustering dendogram

Correlated features are put into clusters using hierarchical Ward-linkage clustering (McCann et al., 2019; Cope et al., 2017). An overview of the resulting dendrogram is shown in Figure C1. A threshold is chosen to make a cut in the dendrogram. This

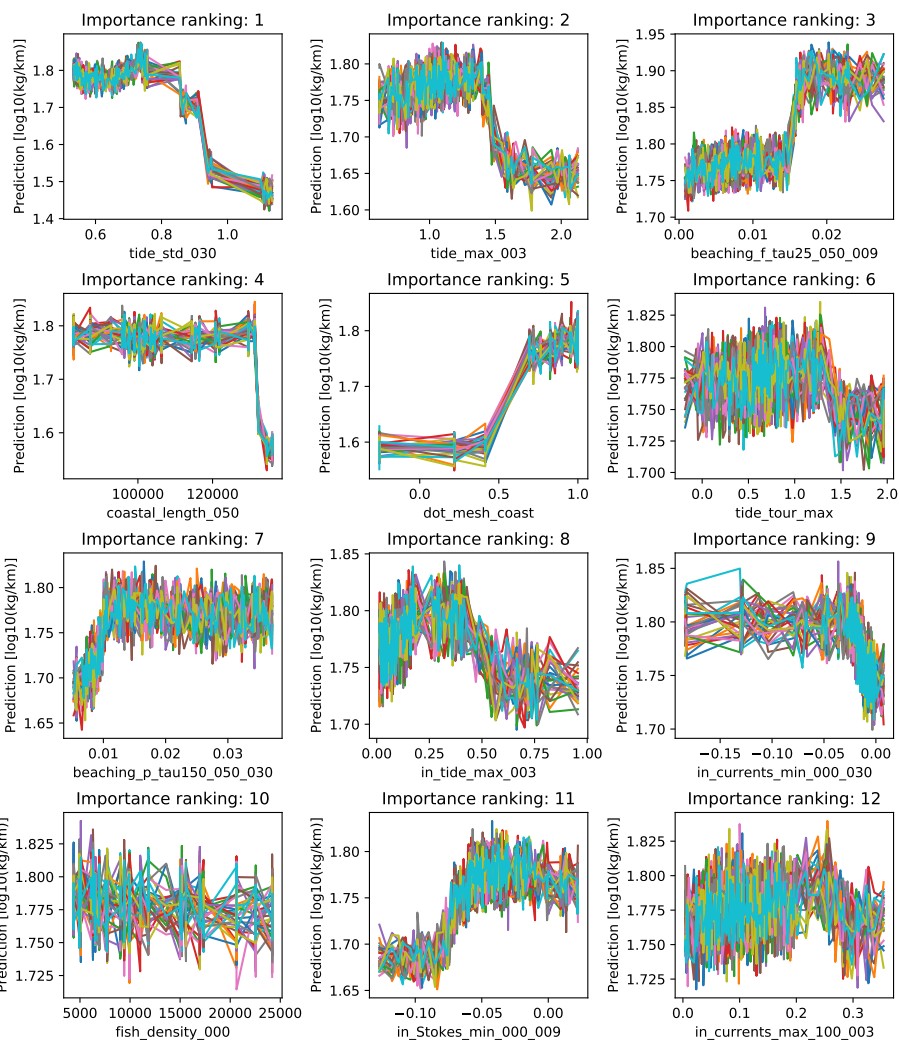

**Figure B7.** Illustrated effect of the 12 most important features (x-axes) on the litter concentrations (y-axes) according to the random forest regression model. For the 12 important features, we vary their value from the minimum to maximum encountered value. All other features are fixed, and some noise is added to illustrate robustness of the relations.

was selected by hand to be a value of 2.3, at which the clusters remain relatively interpretable (e.g. separate clusters for coastal properties and tidal properties). The cut is shown in the figure by the red dashed line. Some general patterns regarding the clusters are indicated in the dendrogram.

**Appendix D: Model pipeline**

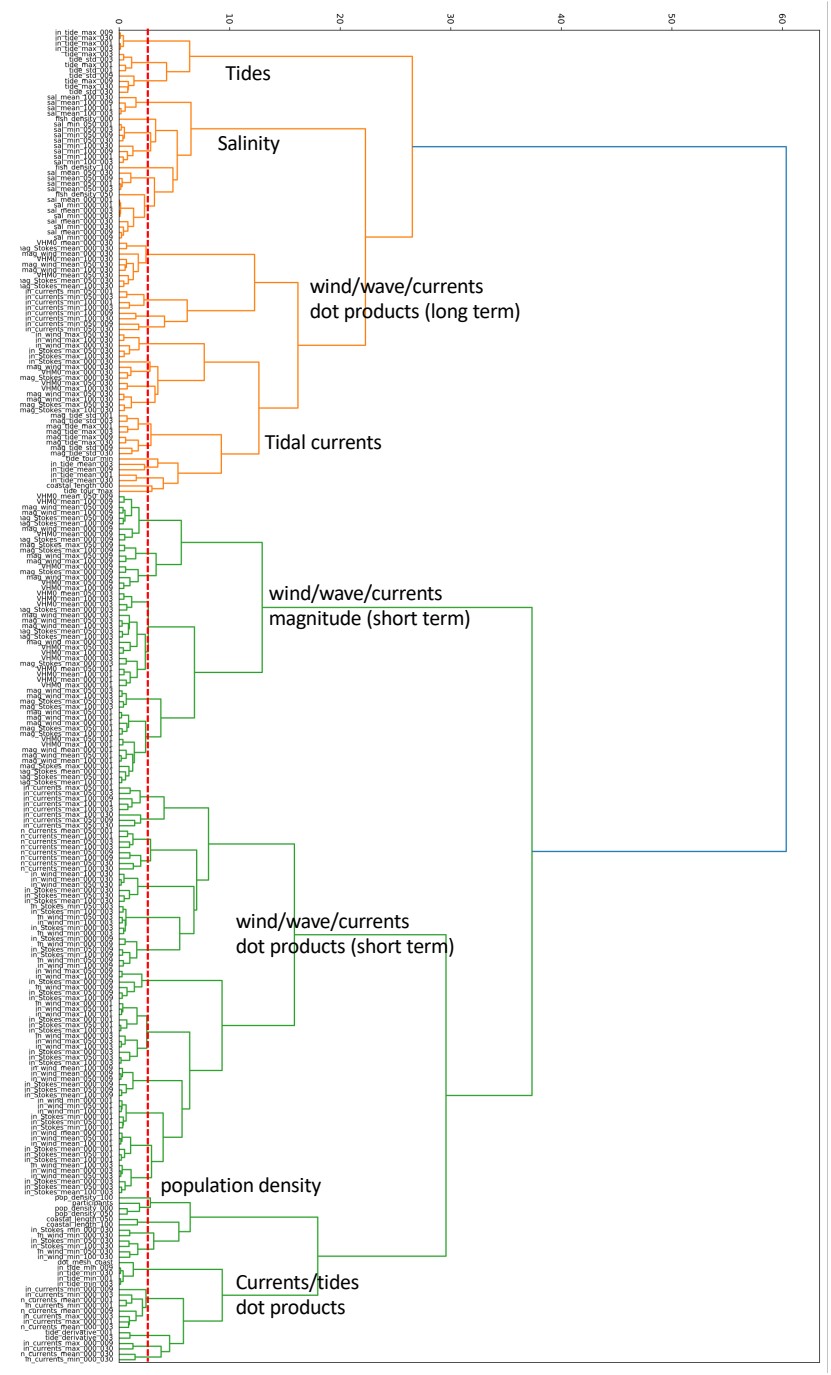

**Figure C1.** Dendrogram used to construct the feature clusters

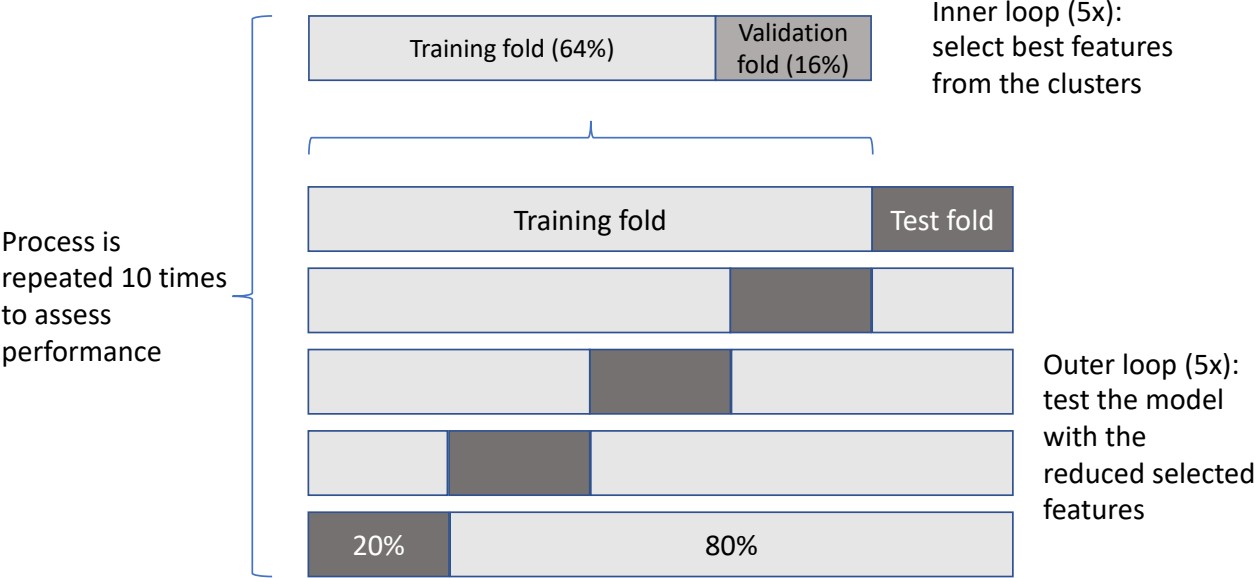

**Figure D1.** Pipeline to train and test the random forest regression model. Nested k-fold cross validation is used to select the best feature from each cluster (inner loop), and to evaluate the model trained with the best features on the test data set (outer loop). The process is repeated to assess the average performance.

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
