# Peer review of "Using machine learning and beach cleanup data to explain litter quantities along the Dutch North Sea coast"

_Ocean Science, 2021_

## Author Response (AR1)

**Summary**

The manuscript uses machine learning techniques to formulate a regression model to investigate the influence of a range of hydrodynamic, atmospheric and coastline geometry factors obtained from publically available data and re-analysis products, as well as particle tracking model runs based on publically available hydrodynamics model output, on the density distribution of (plastic) beach litter on the Dutch coast. The regression model is 'trained' using several years of data from beach cleaning surveys. The main explanatory factors identified are negative correlations with tidal elevation variability and maximum tidal amplitude, and a correlation with coastline orientation relative to residual currents. The particle tracking model runs did not contribute significantly to the predictive capability of the regression model. A trimmed-down version of the regression model, using only the most important contributing factors, was used to generate a 5-year average litter concentration distribution along the Dutch coast.

**General remarks**

This is a very well-written manuscript, that I have read with interest. It mobilises a variety of different data, and uses sophisticated statistical techniques to analyse them together. It addresses a clear need to improve understanding and predictive capability of beaching plastics in the marine environment. Having said that, the manuscript is not completely convincing, and leaves me wondering about a few things:

**We would like to thank the reviewer for this positive feedback on our manuscript, and the thorough remarks and comments which we will discuss below.**

1. By which mechanism do tidal variability and amplitude control the beach litter distribution? Do higher variability and amplitude promote removal of previously beached material and in that way define a lower equilibrium concentration? Or does it simply spread the litter more in the cross-shore direction making it harder to detect in the beach surveys? How might this be illustrated? Or is the preference of the regression model for tidal variability and amplitude an artifact of the method (see also next points)?

We have addressed the different effects that tides might have on litter concentrations in the main text (line 314 track changes)::

In general, a higher tidal maximum and variability lead to less litter measured on the coastline (see the Appendix B5 for further details). A higher tide during or preceding the cleanup could re-suspend some of the litter from the beach. Furthermore, a higher tide encountered during the cleanup stage reduces the beach width that can be sampled. Perhaps a stronger variability in the tidal height leads to less persistent high strandlines where the highest litter concentrations are normally found (Heo et al., 2013). It has been shown in numerical studies that residual tidal currents can lead to a net transport of both suspended and floating matter (Gräwe et al., 2014; Børve et al., 2021; Schulz and Umlauf, 2016). While the regression model indicates that tides play an important role, it is difficult to separate the causal relations between all these different effects and the litter quantities found on beaches. To quantify this in more detail, further experimental and numerical studies are required.

We have added a paragraph in the conclusions and recommendations discussing that further experimental and numerical studies are needed to investigate the effect of tides, since we do not know the exact causal effect (line 475 track changes):

Future studies could further investigate the causal relations between the variables seen as important predictors by the regression model and the litter concentrations found on beaches. This is especially the case for tides, which constitute the two most important features in the regression model (see Figure 5). Experimental studies could further determine whether lower litter concentrations at locations with higher tidal variability are mainly caused by litter resuspending back into the sea, or for example due to the fact that less area of the beach is sampled during high tide. It should additionally be investigated how these effects compare to the role of (residual) tidal currents, as it has been shown that this can play an important role in transporting suspended matter towards the shore (Schulz and Umlauf, 2016). Experimental investigations can be done in combination with numerical studies of the nearshore marine environment, to capture the interactions between processes such as tides, waves, and particle sizes (Alsina et al., 2020).

2. A regression model cannot be used to prove causality. Indeed, the authors acknowledge that many of the factors investigated are not independent, as well as admitting that there are many regression factors compared with the number of available data points. I think that the authors can do more to make their case:

a) they can 'cripple' the regression model by removing (some of) the most important factors (rather than those that do not matter much) and show that the resulting model has significantly reduced predictive capability (as opposed to shifting predictive power to a different factor);

b) they state that the particle tracking model does not add to the predictive capability as it includes most of the factors from others sources incorporated in the regression model. So they can treat the particle tracking model as an independent (causal!) experimentation tool, and validate its results against the field data as well as cross-comparing with the results of the regression model, and subsequently disable the most important processes suggested by the regression model to directly illustrate their influence on the results.

The reviewer is correct that this is indeed a difficult aspect of using data-driven machine learning techniques to make predictions, as opposed to traditional physical models, where it is easier to draw conclusions on causal relations between observations and certain model parameters or processes.

We have included a figure in the supplementary information now (Figure B5), where we checked what happens when leaving out certain important features from the regression model. We have also added (line 545 track changes):

In Figure B5, we analyse the effect of leaving out certain feature categories on the model performance. The random forest can create a highly non-linear map between the features and corresponding response. It is therefore possible that when using a large set of features and leaving out one important explanatory variable, it will use a combination of the remaining features to still obtain a good fit. We therefore only use the top 10 features in this analysis, and exclude the Lagrangian model variables, as these implicitly contain information on the other features. As can be seen, leaving out a certain category of features

reduces the model performance. This can especially be observed when leaving out all features regarding tides, and the two features regarding coastal properties ( $l_{coast}$  and  $n_{grid}$ ). The mean Pearson correlation coefficient decreases and the variance of the model performance increases.

Issue b) would be very interesting to check, but this is out-of-scope for this project. One of the points of this study was to investigate what kind of processes might play an important role near the coastline regarding the beaching of litter. Once we have a better understanding of this, we can include these processes to model the beaching of litter more accurately. As we discuss in the main text (line 148 track changes), most contemporary particle tracking simulations use e-folding time scales to model how quickly litter might end up on the coastlines. These time scales are assumed to be constant for now, which is of course a simplification. Since these particle tracking simulations do not yet contain most of the relevant parameters in the beaching parameterization, we can also not disable them and cross-compare it with the regression model. But this is definitely something which should be implemented and investigated in the future.

We are currently further investigating whether we can indeed combine our particle tracking models with machine learning models, and cross-compare the results, see for example doi.org/10.5194/egusphere-egu21-274.

3. Have all potential causal factors been included in the analysis? The particle tracking results suggested that rivers may be the dominant source of beach litter. One factor I'm missing (and that may be as important or more important than tidal variability) is the proximity of the nearest riverine source (in the upstream direction). Should this not be added as a regression factor? It's possible that in the current setup the influence of a factor like this is attributed to the tidal factors which have a spatial gradient in the same direction.

This is a very good suggestion. We have re-run our model, with salinity as a proxy for the distance to the closest river. By doing this, a low salinity indicates that there is a river close-by in the upstream direction.

We have also re-run our Lagrangian simulations, per the comments we received on a related paper in Geophysical Research Letters (for the preprint see DOI: 10.1002/essoar.10508992.1). In this paper the reviewers noted correctly that the dataset used for the currents already contains the net effect on the ocean flow induced by tides. We therefore removed the additional tidal forcing in our Lagrangian model simulations. This is adjusted in the main text (line 129 track changes):

"We do not add additional tidal forcing to the Lagrangian model (Sterl et al., 2020) since the net effect of tides is already included in the ocean surface current data set (Global Monitoring and Forecasting Center, 2021)."

One of the first things to note, is that more importance is given to the litter fluxes calculated from the Lagrangian model runs. This is good news, as it indicates the Lagrangian model runs have greater predictive power compared to what we saw previously. We now also see that the model attributes more of the plastic pollution to originate from fishing activity and coastlines, which is exactly in line with what we found in our related paper (10.1002/essoar.10508992.1): about 40% of the plastic litter in the Netherlands is observed to originate from the fishing industry. More importance is attributed now to litter fluxes from

**fisheries and coastlines compared to rivers. Furthermore, proximity to rivers, using the salinity variables we introduced, is not seen as an important predictor by our model.**

4. Is the regression model as good as it seems? I would like to see direct comparisons of predicted annual distributions with the observed annual spatial distributions in Appendix A. I'm aware that these data were used for training so it's not a validation, but it would illustrate to what extent the regression model captures the temporal and spatial variations, which are not represented in Figure 4.

We have added figures to appendix A comparing the litter predictions with the litter observations per year per location. We plot the test data points here for each of the outer loops in the 5-fold cross validation process, so technically these points are not contained in the training data. We will use the illustration below (example year 2017) to discuss some of the things that the model gets right, and some of the things that the model is missing.

Firstly, the lower litter concentrations in the south are reproduced well by the model. Furthermore, the model accurately captures the boundary at which litter concentrations strongly increase, marked by a) in Figure R1. North of this boundary, litter concentrations are generally high, which is reproduced well by the model for all years. In the observations the litter concentrations are often slightly higher on the eastern most island (Schiermonnikoog), marked by b) in Figure R1, which is also reproduced quite well by the model for the various years. One thing that the model misses are outliers, of which one example is marked by c) in Figure R1. Exceptionally high litter concentrations can be caused by all kinds of factors, such as very local 'point sources' of litter which are not captured by models, or subgrid-scale processes. We also discussed this subgrid-scale variability using the variogram analysis. Looking at Fig 4 in the manuscript, it can already be seen that the model misses some variability: some outliers can be observed of very low observed litter concentrations (the points in the upper left corner of the scatter plot outside of the  $2\sigma$  lines), and very high observed litter concentrations (the points in the lower right corner of the scatter plot outside of the  $2\sigma$  lines). We have added subsequent discussion to the main text (line 302 track changes):

It can be seen that there are two kinds of outliers in Figure 4: low observed litter concentrations not captured by the model (points in the upper left corner of the scatter plot), and high observed litter concentrations not captured by the model (points in the lower right corner of the scatter plot). This can be explained by the fact that the model is not able to capture all variability contained in the observations. As the hydrodynamic and wind data in the model have a limited resolution, subgrid-scale effects are missing (see Section 4.2). Furthermore, local point sources of litter (both spatially and temporally, e.g. shipping container accidents) are not captured by the model.

I would like to challenge the authors to address these points, which would significantly improve the manuscript, and hence recommend major revisions.

**Specific comments**

1 10-12: recommendations for removal. You currently don't really give any, please remove this sentence from the abstract or work out clear recommendations and state them specifically here.

We have added a specific recommendation to the abstract.

1 77 Some sites were sampled multiple times per year. Please describe how these were treated in the analysis. Would this bias the results and how?

In cases where multiple data points are plotted per year, different stretches of beach were cleaned up. For example, once the northern stretch of beach, once the southern stretch. However, these individual stretches are too small to be plotted on the map, which is why some stages contain multiple points per year. We have added this clarification in the caption as well:

**For stages with multiple data points per year, different stretches of beach were cleaned (e.g. once the northern side, once the southern side).**

Table 1. daily mean currents. This would aliase in a tidal component. What would be the influence on the results? Also, are the bathymetries used to produce the various products used consistent?

The North West Shelf Reanalysis product contains the net effect of 15 tidal constituents. Therefore the net tidal effect is captured in the Lagrangian simulations. These Lagrangian simulations are meant to capture the large-scale particle transport at long time scales. In the statistical analysis we can then use the FES2014 data to calculate the tidal components at high frequency to assess their influence. The Waves reanalysis and ERA reanalysis products use the same bathymetry data (ETOPO2), the North West Shelf reanalysis product uses the GEBCO bathymetry data, the FES2014 data uses a composite, so these might differ slightly. However, what we consider to be more important, is that the atmospheric forcings are consistent (all obtained from the ERA5 data), and that all data are assimilated. We have added text for clarification (line 111 track changes):

High frequency tidal forcing has been used to produce the ocean current data, but output is only provided daily. To capture the effects of tides on a high temporal resolution, FES2014 data are used. Tidal currents ( $U_{\text{tides}}$ ) and heights ( $h_{\text{tide}}$ ) are calculated, taking the  $M_2$ ,  $S_2$ ,  $K_1$ , and  $O_1$  constituents into account (Sterl et al., 2020), as well as the  $M_4$  and  $M_6$  components which have been shown to play an important role in transport of suspended particles in the North Sea (Grawe et al, 2014).

Figure 2 / rivers: why has only a subset of rivers been included? What has driven the decision to include certain rivers but not others? Please include a list of rivers used in an appendix?

Only rivers which transport more than 0.2 tonnes of plastic litter into the ocean according to the Lebreton dataset are plotted in the figure. We have added this as clarification to the caption:

*While all rivers from Lebreton et al. (2017) are included in our analysis, only rivers predicted to transport more than 0.2 tonnes of plastic litter into the ocean are plotted here.*

Please note that we did not produce these riverine data, which are provided by Lebreton et al. at doi:10.6084/m9.figshare.4725541

equation 1. In reality, I would expect tau\_beach=F(x,y,z,t)? Please discuss why a constant was chosen?

This is a good point. One aim of this study is to get a better understanding under what conditions we can expect more litter to beach, since at the moment this is not well understood. This means we also do not yet know how this beaching time scale can be best parameterized. Only after getting a better understanding of the important beaching processes, we can attempt to improve this parameterization. So, in other words, moving from a constant to a more complicated function could be follow-up work to this manuscript. We have added a clarification to the text (line 148 track changes):

While in reality  $\frac{\delta}{\delta}$  might vary significantly both in space and time, it is unknown how this can be best parameterized (Onink et al., 2021). We use the Lagrangian model simulations to capture the large-scale transport of litter, and allow the regression model to pick the most appropriate value for  $\frac{\delta}{\delta}$  allow the regression.

Figure 6 and associated text: it would be helpful to have an indication what the principle components may dominantly represent if that's at all possible?

We have investigated the correlations between the principal components and the features, and added a discussion of this to the text (line 384 track changes):

The first principal component shows the highest absolute correlation (Pearson R: 0.45) with long-term tidal variability (with a lead time of 30 days). The second principal component shows the highest absolute correlation (Pearson R: -0.58) with the nearby coastal length (within a radius of 50km). As the measurements taken between 52--53°N are clustered quite closely together, this indicates that conditions regarding tides and coastline geometry are relatively similar for these locations.

1 67 month: which month of the year?

We have added a clarification to the text:

During this tour, every year in August, the entire Dutch North Sea Coast is cleaned up by volunteers.

**1 82 August: why one month? Why this month?**

See comment above. This is the month that the tour is always organized.

Technical corrections

15. what: which

adjusted

16 remove might

adjusted

1 6 variability: of what? types/size/volume/??

added: beach litter concentrations

1 10 what: which

adjusted

1 15 increase: release?

adjusted

1 15 need of: need for

adjusted

1 21 other sinks: such as?

Changed to: In addition, the plastic concentrations found on beaches are generally higher compared to other environmental compartments such as the surface water or the seafloor...

1 23 ...plastic items by removal, ...

adjusted

125 ... reduction of new plastic waste...

Awareness can also lead to more people picking up old plastic waste, so we leave it as is

**175 weighing devices: scales**

**adjusted**

1 80 mean currents: surface? depth-averaged?

added: mean surface currents

Figure 1. Please include a scale vector for the currents.

A scale vector was added to the figure

Figure 1 caption. Please include references to the data sources. Add a period to the end of the caption.

Data source references were added

1 107 there's a space between models and the period ending the sentence.

adjusted

1 108 the daily-mean ocean-surface currents

1 109-111 include references to the data sources?

Regarding the two comments above: the temporal resolution and references are stated in the table to make the text more readable

1 134 radius of 50km: from where?

Added '...from the coastline...'

Many of the figures: please re-consider the choices of colours to facilitate colour-blind people.

We thank the reviewer for this comment, we weren't aware that the standard quantitative colormap in python is not colorblind friendly. We use a colorblind friendly version now. Also we use greyscale for the rivers in Figure 2 now.

Figure 2: please include axes.

We have added longitude/latitude axes

1 187 what: which

adjusted

1 188: distance: between what?

Added separation distance between the different cleanup locations...

**1 345 less: fewer**

**adjusted**

1 403 insights into which processes may be causing

**adjusted**

1 404 and which length scales should be

The authors present an interesting research article to identify features that favor marine litter deposition. The article is well laid out, and the subject of this article is very relevant to today's society. The outcomes of the article could be used to further optimize the organization of beach cleanups.

The authors present a well laid out overview of the literature and lead well into the research question.

**We would like to thank the reviewer for these very encouraging comments.**

In section 3.2.1., the authors describe, that they create a large set of combinations for the explanatory variables. Later in 3.2.2., features that correlate are then assigned to clusters to reduce the dimensionality of the data. I was wondering, if this could have been solved by creating a medium sized set of combinations in the first place, in order to make the clustering part obsolete?

We chose this approach, since we didn't know a priori which set of features would be the most relevant. We didn't know a priori what kind of lead time and what kind of radius of influence is the most dominant in causing litter to beach: does the amount of litter that is found on the beach mainly depend on longer time scales, which might be slowly leading to an increasing standing stock of litter, or is most of the litter arriving on the beaches in the previous couple of days? Since readers who would want to implement similar approaches would also face the same questions, we find it informative to keep the full analysis in the manuscript. We have added some clarification to this reasoning in the manuscript (line 270 track changes):

Some features correlate as these are, for example, derived from the same variable, but for a different radius or lead time. However, we do not know a priori which of these radii and lead times are the most appropriate predictors for the beached litter quantities. For example, litter concentrations might be influenced by long-term processes, slowly increasing the standing stock of litter on the beach, or the concentrations could be better predicted by conditions on the day leading up to the cleanup stage. Since we do not know this, we let the algorithm select the most appropriate variables.

In section 3.2.2. it is not quite clear to me, if the test data is at some point used as training data during the Nested 5-fold cross validation training process. The authors should describe this in a little more detail.

The k-fold cross validation procedure is also illustrated in Figure D1. One divides the available dataset into a part for training and a part for testing. The test data is never used for training within the outer loop. This means that the model performance is not overpredicted. In fact, it may be slightly underpredicted because the model becomes more accurate as more data becomes available, so one could even say that the real performance of the model could be slightly higher. This procedure is quite standard, we've added a link to the book by Hastie and Tibshirani in which this method is explained more clearly for the interested reader. We have added a clarification to the manuscript (line 280 track changes):

In the inner loop, 80% of the training data (i.e. 64\% of the total data) is used to train the model, and 20% of the training data (i.e. 16% of the total data) is used to calculate the importance of the features, also repeated 5 times. Since in the inner loop none of the test data

are used to train the model, we do not overpredict the model performance (Hastie et al., 2008)

Could the authors explain more in detail why they chose Random Forest as regression algorithm compared with other regression algorithms?

We have added more clarification to the text (line 259 track changes):

This model allows us to capture non-linear relations between the features and response. It is a non-parametric model, and does not require prior knowledge on the model structure. These are both important reasons to choose the specific algorithm: coastal processes affecting dispersion of marine litter are highly complex (Sebille et al., 2020), so we do not know a priori how the different environmental variables might interact, and how non-linear these interactions might be. The random forest regression model can aid in scientific knowledge discovery (Bortnik and Camporeale, 2021): it gives us Gini importances for all features (Nembrini et al., 2018). This is another reason for choosing this specific algorithm, as it provides us information what processes are important for predicting beached litter concentrations.

In section 4.3., it was not crystal clear to me, why a new regression model with the Top 8 features was trained compared with using the model that was already trained. Could the authors explain the reasons behind this?

We have added clarification for the reasoning in the manuscript (line 421 track changes):

We choose to use a model trained using the top 8 features only for the extrapolations, as increasing the amount of features does not increase the predictive performance (see Figure B5). Furthermore, reducing the amount of features simplifies the computations, as we do not need to compute all 391 variables again for the entire Dutch coastline.

In section 5., the authors give an extrapolation of how much litter is located on the Dutch North Sea coastlines. Can these extrapolations reliably be extrapolated to the whole coastlines of Europe etc.?

No, the model is trained using conditions in the Netherlands, and should therefore not be used to make extrapolations for other regions. Machine learning models generally perform poorly when making extrapolations for conditions not seen before in the training data. Conditions might be very different for other regions (e.g. much stronger currents, more complex coastline geometries, different coastline substrates). We have added a clarification regarding this to the manuscript (line 484 track changes):

The model itself can not directly be used for other geographic regions, since the features used to train the algorithm are specific to the region of interest. The model is likely to perform poorly when making extrapolations for conditions not present in the training data. As an example, the substrate of beaches is likely to have a large impact on litter concentrations (Hardesty et al., 2017), which are relatively uniform in this analysis (all sandy beaches).

I was missing some context of the extrapolations (16T - 31T), i.e., how much waste collect cleanup missions during a whole year. Could the authors include this information?

This is a good point. We have added a table to the supplementary information, where one can see that the total weight gathered over the various years varies from 9,872 to 20,078 kilograms. See table A1

We have added a small discussion on these numbers to the manuscript (line 357 track changes):

The total amount of litter gathered during the cleanup campaigns, and the total amount of kilometers sampled per year is presented in Table A1. The total amount of litter gathered varies from 9,872 to 20,078 kilograms. This is in line with the expected total amount of litter predicted by the model, since the majority of the coastline (222 to 262 kilometers out of 365 kilometers) was cleaned up during the cleanup campaigns.

**Summary and scientific relevance:**

The purpose of this manuscript is to investigate the physical processes leading to the accumulation of litter on the beaches of the Dutch North Sea coast. Data from six years of beach cleanups were used to fit a variety of environmental parameters with a random forest model to identify possible correlating variables affecting litter accumulation on the coast. Tidal height and variability were found to be the strongest explanatory variables, leading to a decrease in litter accumulation at the coast with increasing tidal height and variability. In addition, shoreline geometry was found to have explanatory power for litter accumulation on Dutch beaches. Based on the best explanatory variables, the authors extrapolated the distribution and abundance of beached litter on the Dutch North Sea coast, which may contribute to effective cleanup strategies by identifying hotspots of litter accumulation along the coast.

**General comments:**

The manuscript is well organized and generally easy to follow. The presentation of data and methods is very well structured and the results are clearly illustrated. The method presented for studying coastal litter accumulation is very innovative and can help provide valuable insights into the processes governing the beaching of marine litter in shelf seas. I wonder about two points that I think are particularly important for coastal shelf seas. I would like to recommend this manuscript for publication after moderate revision according to the following points.

We would like to thank the reviewer for these kind words, and the very useful comments below.

• As far as I understand it correctly, the authors of this study used the AMM7 model of the Copernicus Marine Service for the advection of the virtual particles. One key point I wonder about is how the authors analyzed the influence of tides on marine litter washing ashore on Dutch beaches. The authors found that tidal height is the most important influencing variable, resulting in a decrease in the amount of litter as the variability and height of the tides increases. Numerical studies (e.g., Stanev and Ricker, 2020) and observational studies (e.g., Meyerjürgens et al., 2020) have found that tidal forces (including overtides) have a significant influence on the length of Lagrangian trajectories and particle residence times, in addition to affecting particle dispersion at different spatial scales. Since the authors used daily average values of ocean current fields, the effect of tides is suppressed in the analyzed Lagrangian model, which is very important for the reader to classify the results of this study. Please clarify and discuss the important effects of tidal currents on the results of this study.

We use the CMEMS North West Shelf Reanalysis product in our analysis for the ocean surface currents. This product uses 15 tidal constituents, so their net effect is contained in the

Lagrangian particle simulations (i.e. the long-term effect on time scales above a day). In our first submission we included additional tidal forcing in the Lagrangian particle simulations as well, using the FES2014 data, at a high temporal resolution (the integration time step in the Lagrangian model is 20 minutes). But, as reviewers noted correctly in a related paper we submitted (van Duinen et al. (2021), 10.1002/essoar.10508992.1) this means the effect of tides is overpredicted. We therefore re-did our simulations without this extra tidal forcing.

To still include the high frequency effect of tides on our analysis, we included the tidal height as a separate feature. Since this is obtained from the FES2014 data (which is spectral data), we can calculate the tidal components at any frequency we want. This allows us to calculate the tidal height and currents at any given lead time, or what the maximum tidal height was during the cleanup stage. Given the feedback below, we have now extended the previously included components (M2, S2, K1, and O1) with the M4 and M6 components.

We have added some comments on the different effects that tides might have on litter concentrations in the main text (line 314 track changes):

In general, a higher tidal maximum and variability lead to less litter measured on the coastline (see the Appendix B5 for further details). A higher tide during or preceding the cleanup could re-suspend some of the litter from the beach. Furthermore, a higher tide encountered during the cleanup stage reduces the beach width that can be sampled. Perhaps a stronger variability in the tidal height leads to less persistent high strandlines where the highest litter concentrations are normally found (Heo et al., 2013). It has been shown in numerical studies that residual tidal currents can lead to a net transport of both suspended and floating matter (Gräwe et al., 2014; Børve et al., 2021; Schulz and Umlauf, 2016). While the regression model indicates that tides play an important role, it is difficult to separate the causal relations between all these different effects and the litter quantities found on beaches. To quantify this in more detail, further experimental and numerical studies are required.

Since this is an important aspect, we have added this as a recommendation for future research to the conclusions and discussions (line 475 track changes):

Future studies could further investigate the causal relations between the variables seen as important predictors by the regression model and the litter concentrations found on beaches. This is especially the case for tides, which constitute the two most important features in the regression model (see Figure 5). Experimental studies could further determine whether lower litter concentrations at locations with higher tidal variability are mainly caused by litter resuspending back into the sea, or for example due to the fact that less area of the beach is sampled during high tide. It should additionally be investigated how these effects compare to the role of (residual) tidal currents, as it has been shown that this can play an important role in transporting suspended matter towards the shore (Schulz and Umlauf, 2016). Experimental investigations can be done in combination with numerical studies of the nearshore marine environment, to capture the interactions between processes such as tides, waves, and particle sizes (Alsina et al., 2020).

• Another point I wonder about is why the authors of the study use anthropogenic variables such as fishing intensity as possible sources (for the Lagrangian model), but do not include anthropogenic factors (other than population density) in the statistical analysis. I would recommend that the authors include anthropogenic parameters such as "ship density" and "fishing activities" (which can be taken from EMODNET, for example) as possible explanatory variables in their analysis to get a sense of how anthropogenic factors affect coastal litter accumulation compared to hydrodynamic and atmospheric parameters.

We have added more anthropogenic factors to the set of features to make this consistent. The population within a given radius was already included ( $n_{pop.}$ ), we have added the fishing effort in a nearby radius as well now ( $n_{fis.}$ ). Furthermore, we have included salinity (*sal.*), a parameter which can be used as a proxy to see how close a given beach is located near upstream river mouths. The entire analysis was redone including these features.

Our results now show that fisheries likely play a large role in the amount of litter that is found on the Dutch coastlines. The Lagrangian model run with particles released at fishing locations is now one of the most important explanatory features, ranked 3rd. In general, the Lagrangian model simulations now play a more important role compared to our first submission. This can be explained due to the fact that the new Lagrangian model runs are likely more accurate now, as in the first submission the effect of tides on the particle trajectories was overestimated. Additionally, nearby fishing intensity ( $n_{fis.}$ ) is ranked as the 10th most important feature.

We have not included ship density as a separate variable, as a lot of the litter on beaches is estimated to originate from the fishing industry specifically (see .e.g van Duinen et al. (2021), 10.1002/essoar.10508992.1). Littering from ships in the North Sea is highly regulated, and likely more of a very local nature in space and time (e.g. container accidents doi.org/10.3389/fmars.2021.607203). This is difficult to include in the model without having a priori knowledge of when and where this happened, which is out-of-scope for this project.

**Specific comments:**

Line 110: The authors have considered the most important tidal components in their calculations. In the North Sea, shallow-water tides (M6 and M4) play an important role in the currents and advection of particles. Please keep this in mind in your analysis and consider it when discussing your results.

We thank the reviewer for this comment. We have added the M4 and M6 tides to our analysis, as we indeed found in the literature that these components play an important role in the transport of suspended particles. We have added to the manuscript (line 113 track changes):

Tidal currents ( $U_{\text{tides}}$ ) and heights ( $h_{\text{tide}}$ ) are calculated, taking the  $M_2$ ,  $S_2$ ,  $K_1$ , and  $O_1$  constituents into account (Sterl et al., 2020), as well as the  $M_4$  and  $M_6$  components which have been shown to play an important role in transport of suspended particles in the North Sea (Grawe et al, 2014).

Line 143-144: Unclear to me. Please rephrase this.

We have rephrased this part (line 153 track changes):

Each virtual particle starts with a unit mass. Each time step that a virtual particle spends near the coast, a fraction of its mass is lost due to the beaching process. This means that as  $t_{coast}$  increases for a virtual particle, a fraction of its mass is lost, which is calculated using \eqref{eq:beaching}. For each virtual particle, we calculate where and when it loses mass due to the beaching process. These masses lost to beaching are binned in a  $1/9^{\circ} \times 1/15^{\circ}$ beaching flux histogram for each day.

Line 174-175: I can't see a dashed brown line in Fig. 3a. Do you mean the brown arrow in this context?

Yes, this should indeed be an arrow, we have adjusted the text.

Line 213: Please clarify why Utide is not considered as a scalar feature.

We have added the tidal currents as scalar and directional features now, as this was indeed not consistent. At first, they were left out since the tidal height was already included, but we added them now since it might be important to consider the currents to include the effect of residual tidal transport. As can be seen in the main text, one of the important explanatory variables is now the dot product of the tidal velocity with respect to the coastline.

Line 344: This sentence seems incomplete.

We have modified the sentence a bit for more clarity (line 395 track changes):

For the smallest lag distance (h = 5 pm 5 mm), we find  $hat_{gamma} = 0.08$ . This variance estimate was also used to create the error bars in Figure 4.

Line 352: Where does this grid size come from? AMM7 should have a resolution of 7 km x 7 km. Perhaps it arises from the inclusion of lower resolution Stokes drift data in the numerical grid? This is not clear to me. Please clarify and add a section in Section 3.1.1 on how your grid size is defined by merging the different data sources.

The different sources of data have different grids. We have clarified this in the text now (line 404 track changes):

The grid sizes used for our numerical data ranges from about 7 km (the surface current data), to about 20 km (the wind data). This means that the variance at and below these length scales is not captured by the numerical data. The variance calculated for lag distances up to 20 km is quite substantial ( $\lambda = 0.05-0.12$ ).

**Technical corrections:**

Line 13: ... the Dutch coastline.

We consider the Dutch North Sea coastline here, inland waters and shorelines around the Wadden Sea are not taken into account

Line 15: ... the need for...

adjusted

Line 75: ...end of the stage...

adjusted

Line 76: Most litter found... or Most of the litter found...

adjusted

Line 152: ... on how...

adjusted

Line 184: ... that are...

we are not sure what is meant here, the sentence seems fine to us

Line 218: ...are calculated...

adjusted

Line 225: ... is used...

adjusted

Line 227: ... is available.

adjusted

Line 236: ... from the coastal population.

adjusted

Line 323: "not to have" not well formulated, please rephrase.

Changed to

It is estimated that the number of participants taking part in the tour does not have a large influence on the amount of litter that is found

Line 363: ... coastline into...

adjusted

Line 381: ...play an important role...

adjusted

Line 386: ... is seen...

adjusted

Line 408: ... the importance...

adjusted

Line 420: ... can be taken into account....

adjusted

---

## Author Response (AR2)

Review of ' Using machine learning and beach cleanup data to explain litter quantities along the Dutch North Sea coast ' by Kaandorp et al.

I have read the responses of the authors and the revised manuscript, and find it significantly improved. I have mostly minor suggestions (see below), except for one point that may require some further work:

We would like to thank the reviewer for going through our manuscript once again, and for this positive feedback.

The authors have used a range of lead times in the scenario simulations, with a maximum of 30 days. Yet, obviously, such a time period is too short for litter released further away to reach the area of interest. Also, the most important factor identified (Figure 5) has a lead time of 30 days. This suggests that longer lead times may (also) be important, or possibly even more important. I would suggest additional simulations, with progressively longer lead times, until increasing the lead time does not add additional cases to, say, the top 10?

We understand where this comment comes from, but want to highlight that additional simulations are not necessary because of several reasons:

- Firstly, and most importantly, there is in fact information in the model on litter from further away reaching our area of interest. In the Lagrangian model runs, particles are released at locations around the European shelf where we expect litter to enter the sea. These particles are tracked for at least 2 years. To be more specific, we looked at the maximum beaching time scale ($\tau_{beach} = 150\ days$), and advect the particles until they have lost more than 99% of their original mass due to the beaching process (which is after spending 691 days next to the coast, or 1.9 years). We acknowledge that this was perhaps not clear enough in the text. We have added (l.152 track changes):

  *Particles are tracked until they have lost more than 99% of their initial mass in the most conservative scenario of $\tau_{beach} = 150\ days$. This means that particles are deleted when they have spent more than 691 days near the coast.*

  And, for further clarification (l.259 track changes):

  *One benefit of adding beached litter fluxes from the Lagrangian particle simulations, is that potential sources of litter far away from the beaching location can be included. While the radius of influence for all features goes up to 100 kilometers, the Lagrangian model features can still include information from further away, since the virtual particles are tracked indefinitely as explained in Section 3.1.2*

- Secondly, the most important factor identified is indeed related to a lead time of 30 days, namely the tidal variability within 30 days. However, tides are periodic, and with a period of 30 days we are able to capture the most important variability up to the spring-neap cycle. Extending the period of time would not add much more information to the model.

l. 31. influence: please specify in which way(s)?

*Added '..., with more litter accumulating in areas with increased backshore vegetation.'*

l. 40. move 'e.g.' to before the first reference

Removed the e.g.

l. 53. we will build

Adjusted

l. 83. 'all averaged over August': this carries the implicit assumption that the plastic that was found beached recently. Is that true? See also lead times remark above.

Previous studies (e.g. Ryan et al. (2014), Eriksson et al. (2013)) have shown that the litter turn-over rates are much faster than time scales of more than a month. When beaches were cleaned daily for one week (Ryan et al., 2014), this yielded 2-3 the amount of litter compared to cleaning once a week. Compared to cleaning once a month, it yielded an order of magnitude more litter (Eriksson et al., 2013). This means that with the time scales that we consider, our model should be able to capture most of the accumulated litter.

Figure 1, caption: mean surface currents

Added 'surface'

Table 1. Please add, for each variable/data set, the time period used (August?? Don't think so, but I'm not sure now...). Explain these choices in Section 3.1.1.

We have added additional clarifications to the table caption:

*For variables with an asterisk (\*) data are used from July up to September 2014-2019. For data with a double asterisk (\*\*) data are used for all months from January 2011 up to September 2019, as these are used for the Lagrangian model simulations as well.*

l. 218. Lead times. See above. Why not more than 30 days?

See first response, and response to l. 83. Also, we have added further clarification to the text (l.224 track changes):

*For lead times, we will consider 1, 3, 9, and 30 days. As shown in Eriksson et al. (2013) and Ryan et al. (2014), the turnover of litter on beaches generally happens within time scales of days, meaning that with this range of lead times we should be able to capture most of the litter accumulation. Furthermore, a lead time of 30 days also captures all tidal variability up to and including the spring-neap cycle.*

l. 235. We use salinity (S) as a proxy...

Adjusted

Table 2. Use S for salinity as a header instead of sal.

Adjusted

l 260. what: which

Adjusted

l. 287. Correlation coefficient: was this calculated on the 'raw' data or on the log-transformed data? If the latter, I'm not sure if this should be called 'reasonable' correspondence as log-transformation imposes a strong bias on (perceived) correlations?

This is calculated over the log-transformed data, as is usually done when comparing (plastic) concentrations with observational data, as we are just as interested in variations in low litter concentrations as for high litter concentrations. Given the estimated error bounds from the variogram analysis, and the fact that 94% of the data fall within these estimated $\pm2\sigma$ confidence bounds, we argue that there is a reasonable correspondence.

l. 306. 30 days lead time: here it is again...
See first response, and response to l. 83

l. 332-334. You can only draw this conclusion if the relative magnitude of these sources is realistically implemented in the model in terms of numbers of particles released. I'm not sure if this was the case (or if we know enough about sources to do this anyway)?
We acknowledge that this was maybe too strongly worded, we changed this to (l.346 track changes):

*This could indicate that transport of litter through the marine environment is important to take into account, as opposed to only considering local terrestrial sources.*

l. 373, l. 420, l. 421: what: which
Adjusted in line 420 (for 'variables', since there is a limited number of these), but still use 'what' for 'spatial variability' and for 'length scales'

Figure C1. The fonts are too small to read at 100% magnification
We are sorry about this, but we are not able to make this easily readable at the standard magnification with the many features that we have. The figure is exported as a pdf to enable zooming and close inspection of the figure for the interested reader.